# Solving multi-scenario hybrid flow shop scheduling problem based on an improved probe machine model

**Xiang Tian**[1]*, **Yang Kong**[1], **Xiyu Liu**[2,3]

1 School of Health Management, Binzhou Medical University, Yantai, Shandong, China, 2 School of Business, Shandong Normal University, Jinan, Shandong, China, 3 Academy of Management Science, Shandong Normal University, Jinan, Shandong, China

* tianxiang08@163.com

## Abstract

The hybrid flow-shop scheduling problem is widely present and applied in industries such as production, manufacturing, transportation, and aerospace. In recent years, due to the advantages of nonlinear access and fully parallel processing, the probe machine has shown powerful computing capabilities and promising applications in solving various combinatorial optimization problems. This work firstly proposes an Improved Probe Machine with Multi-Level Probe Operations (IPMMPO) and ingeniously designs general data libraries and probe libraries tailored for multi-scenario HFS problems, including HFS with identical parallel machines and HFS with unrelated parallel machines, no-wait scenario, and standard scenario. Secondly, based on the data libraries of the IPMMPO, two tuple sets suitable for constraint programming modeling are further designed as data preprocessing. Next, a CP model (IPMMPO-CP) applicable to multi-scenario HFS problems is proposed. Finally, based on a large number of instances and real cases, IPMMPO-CP is compared with 9 representative algorithms and 2 latest CP models. The results demonstrate that the proposed IPMMPO-CP outperforms the compared algorithms and models.

## 1 Introduction

Nowadays, the hybrid flow-shop scheduling (HFS) problem, which is widely existing and applied in the steel [1,2], textile [3], glass manufacturing [4] and plastic manufacturing [5] industries, can be traced back to the flow-shop scheduling (FS) problem proposed by [6]. The original FS problem is to consider only one machine at each processing stage of the job. In other words, the number of processing stages in the FS problem is strictly equal to the total number of machines. Different jobs have the same operation sequence, which makes the job sequence the only problem that FS needs to solve. This production method is obviously difficult to adapt to the needs of today's large-scale flexible process and parallel manufacturing. In contrast, the HFS problem considers at least two parallel machines to choose from in at least one of the processing stages. This forces HFS to solve machine allocation and job sequencing problems simultaneously at each processing stage [7]. Compared with FS, HFS can better

**Funding:** This work was supported by the Yantai Social Science Planning Research Project (YTSK2025-197 to XT), the Social Science Plannig Project of Shandong Province (22CGLJ01 to YK), the National Natural Science Foundation of China (72274023 to YK), and the Natural Science Foundation of Shandong Province (ZR2022MG037 to YK).

**Competing interests:** The authors have declared that no competing interests exist.

relieve bottleneck constraints and improve the production capacity and efficiency of enterprises. Since the HFS problem is flexible with the machines available over each processing stage, it is also called a flexible flow-shop [8] or multiprocessor flow-shop [9].

According to the type of parallel machine in the processing stage, it can be roughly divided into two categories: HFS with identical parallel machines (HFS-IPM) and HFS with unrelated parallel machines (HFS-UPM) [10]. The first means that in each stage, the processing time of the same job on all available parallel machines is equal, i.e., the processing time of each stage depends only on the job itself. The second means that in each stage, the processing time of the same job on any two parallel machines is independent of each other, that is, the processing time of each stage depends on both the job and the machine. The HFS problem involving only two stages has been shown to be NP (non-deterministic polynomial time) -hard [11]. Approaches to solving HFS problems are roughly divided into two categories: exact methods and approximate methods.

Exact methods mainly include mixed integer programming (MIP), Lagrangi-an relaxation (LR) method, decomposition method and branch and bound (B&B) method, constraint programming (CP), etc. [12] proposed a B&B algorithm based on energetic inference and global operations to solve the HFS problem. [13] used the B&B algorithm to solve the two-stage HFS problem aiming at the shortest total delay time. [14] developed an LR algorithm incorporating a speed-up strategy for solving the HFS problem with limited buffer capacity. More researchers tend to develop MIP models and solve them with the help of the commercial solver. [15] developed four different MIP models and compared their performance. [16] proposed a MIP model for the reentrant HFS problem and solved it using CPLEX. [17] proposed a MIP model to solve the complex HFS problem arising from a bio-process industry, which took into account constraints such as limited waiting time between processing stages. Represented by MIP, exact algorithms can obtain optimal solutions for small-scale problems, but their solving time also increases exponentially with problem size. Therefore, it is usually only suitable for solving small-scale problems. Based on the viewpoints of existing research [15,16,18], even a reasonably good MIP model can only ensure that the size of the HFS problem where it is effectively solved is within 15 jobs and 5 stages.

Approximation methods can be further subdivided into heuristics and meta-heuristics, among which heuristics are mainly based on dispatching rules and local search. [19] adopted the shortest processing time and first available machine rules to solve the scheduling problem of computer systems. There are also the well-known Nawaz-Enscore-Ham (NEH) [20] and longest processing time [5]. [21] used a modified NEH to generate high-quality initial populations for the discrete artificial bee colony (DABC) algorithm. In addition, there are local search heuristics that aim to tune critical operations on the critical path, such as variable neighbourhood search [22], iterated local search [23], iterated greedy (IG) [9], etc. For the distributed assembly shop hybrid flow shop (HFS) problem with dual resource constraints, some scholars have proposed a mixed-integer linear programming (MILP) model and a knowledge-based iterative greedy (KBIG) algorithm, aiming to minimize the total tardiness [24]. For the HFS problem with consistent sublots and dual objectives of minimizing the makespan and the total number of sublots, Zhang et al. [25] formulated a MIP model and proposed an automatic algorithm design approach to solve it. Bio-inspired meta-heuristics incorporating random diversity search have become a hotspot for efficiently solving scheduling problems in recent years. The more mature swarm intelligence meta-heuristics and their improved versions include particle swarm optimization (PSO) algorithms [4], artificial bee colony algorithms [1,21,26], simulated annealing (SA) [27,28], genetic algorithms (GA) [29], tabu search (TS) [7,30] and ant colony optimization (ACO) [31], etc. However, a single meta-heuristic has more or less defects such as premature or local convergence. Therefore, more

and more researches tend to organically combine different intelligent algorithms to make up for their own shortcomings, thereby improving search efficiency. [32] fused variable-depth search with SA for the multi-stage HFS problem. [33] fused GA and ACO to solve the HFS problem with time windows. [34] designed a hybrid algorithm combining TS local search and PSO to solve the batch HFS problem with setup times. [35] established a population communication mechanism between adaptive GA and discretized PSO to resist local convergence and improve search diversity. The advantage of these approximation methods is that they can construct a large number of effective solutions in a relatively short time, but their disadvantages are also obvious, that is, it is difficult to guarantee the optimality of the solutions [36]. [37] proposed an improved discrete particle swarm optimization algorithm to solve the permutation flow-shop scheduling problem. [38] proposed a wireless network GA to address a multi-objective FS system.

The probe machine (PM), proposed by [39], is the most advanced DNA computing model so far. This model breaks the two mechanisms that limit the computing power of Turing machine (TM), that is, the linear data access mode and the serial processing mode, and replaces it with the non-linear data access and the completely parallel processing mode. The linear data access mode of TM is 1D, while the nonlinear data access mode of PM is 3D [39], which makes PM show strong computing power and application prospects in dealing with problems related to graph theory.

[40,41] not only proved that TM is a special case of PM, but also explored the use of PM theory and DNA self-assembly technology to solve the graph vertex coloring problem. [42] constructed a molecular probe device based on DNA molecular computing, which has application prospects in the fields of biological imaging and disease diagnosis. Tian et al [43] solves the urban light rail route planning problem based on PM. The team of professor Yin studied the application of PM to the maximum clique problem [44], the satisfiability problem [45], and the Chinese postman problem [46]. [47,48] reported the application research of PM in the field of model checking. [49] reported the computational power of connective PM on the shortest path problem. Professor Ma's team explored the application of PM to problems such as the traveling salesman problem [50,51] and vehicle routing with capacity constraints [52]. The above researches show that the computing performance of PM in terms of high flexibility and low complexity is far better than that of TM. However, the original PM only supports single-level probe operation, which restricts the computing power of PM in dealing with complex graph theory problems. In addition, as of now, no research on the application of PM in the field of shop scheduling has been found.

Since 2016, with the development of software and hardware technology, the CP modeling method has received more and more attention as an exact method to solve various shop scheduling problems. IBM CP Optimizer is commonly used in academia and industry to implement efficient CP modeling for scheduling problems. IBM CP Optimizer is a software library that efficiently models and solves combinatorial optimization problems [67]. It uses efficient constraint propagation algorithm and domain reduction technology as its search and reasoning engine, and supports multi-core, multi-thread parallel optimization technology. CP methods have a proprietary optimization programming language (OPL) that combines specialized variables, constraints, and keywords to efficiently express complex spatiotemporal constraints to build compact scheduling models. CP methods have been proven to be successfully applied in various scheduling fields, including but not limited to flexible job shop scheduling problem with parallel batch machines (FJSP-PBM) [57], parallel machine scheduling problem with sequence dependent setup times (SDST) [58,59], team orienteering problem [60], job shop scheduling problem (JSSP) based on DNA algorithm [61], parallel machine

flexible resource scheduling (PMFRS) [62], robot scheduling in retirement home environment [63], and distributed FJSP [64]. Focusing on the characteristics of the hybrid flexible flow shop such as setup time and due dates, [56] proposed a CP approach with the objective of minimizing the total tardiness. Considering the total weighted tardiness objective of the job shop scheduling problem, [55] constructed three models using the MIP and CP methods. The experimental results show that the CP method involves a smaller number of variables and is more efficient to solve. [54] proposed a new CP model for large-sized scheduling problems in multi-product, multi-stage batch plants and verified the effectiveness of the model. [53] transformed the actual online print shop scheduling problem into FJSP with sequential flexibility and proposed the MIP and CP models with the objective of minimizing makespan. The CP methods have been proved to be effective in solving medium-scale and even large-scale scheduling problems. Not only that, CP methods can even prove that the best feasible solution found is optimal when the search automatically terminates [59,64].

In recent years, CP methods have also made some new progress in solving HFS problems. [70] developed a bi-objective CP model with makespan and total energy consumption as optimization criteria by adjusting machines to different speed levels. But this study only addressed the HFS-IPM problem. [73] established MIP and CP models for different types of scheduling problems and conducted computational evaluations. The HFS problem involved also only discussed the HFS-IPM type, and the CP model of this problem does not use "`alternative`" constraints. For the HFS-UPM type, [71] proposed a CP model for the no-wait HFS problem, which does not use "`cumulFunction`" constraints. [72] developed corresponding CP models for HFS-UPM and its several extensions, making outstanding contributions to exploring the application of CP in HFS problems. However, the excessive number of constraints in the developed CP model means that higher computational costs may be required when solving problem instances [71].

Based on the above background, this work first proposes an improved probe machine with multi-level probe operations (IPMMPO). Then the HFS problem is transformed into a graph problem by disjunctive graph, and the data library and probe library are cleverly designed to solve it. Second, based on the two data libraries designed in IPMMPO, this work constructs two sets of tuples dedicated to CP modeling. Based on the above two sets of tuples, a unified CP model (IPMMPO-CP) is established for HFS-IPM and HFS-UPM problems. The contributions of this work are summarized as below.

1. This work explores the application research of the probe machine in the field of scheduling for the first time. For two types of HFS problems (HFS-IPM and HFS-UPM), an improved probe machine model (IPMMPO) that allows multi-level probe operations is designed. The complexity analysis of IPMMPO model is given. a constraint programming model (IPMMPO-CP) is further constructed based on the data libraries in IPMMPO.

2. Based on a large number of instances and real cases, Sect 5 first investigates the performance comparison between IPMMPO-CP and nine representative heuristic algorithms under the HFS-IPM scenario. Sect 6 further explores the application comparison between IPMMPO-CP and two latest CP models under both the standard and no-wait scenarios of the HFS-UPM type. The computational results demonstrate that the proposed IPMMPO-CP outperforms the compared algorithms and models.

The framework for the remainder of this work is organized as follows. Sect 2 briefly outlines the preparatory knowledge involved in this work, including the description of the HFS problem and the introduction of the probe machine model. Sect 3 introduces the proposed

IPMMPO theoretical model in detail, including its data libraries, probe libraries, model structure and complexity analysis, etc. CP modeling based on IPMMPO can be found in Sect 4. Comparative experiments based on a large number of instances, and application comparisons in real-world cases are given in Sect 5 and Sect 6, respectively. Sect 7 summarizes this work.

## 2 Preliminary knowledge

### 2.1 Problem description

The HFS problem can be described as follows. $n$ jobs need to be processed in $s$ stages, in which each job must be processed on stage 1 first, then on stage 2, and so on, and the last step is processed on stage $s$. It is required that in $s$ stages, at least one stage has more than one machine available. As shown in Fig 1, $M_{k,m_k}$ $(1 \leq k \leq s)$ represents one available machine in stage $k$, where $m_k$ represents the total number of available machines in this stage. As mentioned earlier, according to the type of parallel machines in the processing stage, there are two types of problems, HFS-IPM and HFS-UPM. It can also be found from Fig 1 that when only one machine is available in each stage, the HFS problem reduces to the regular flow-shop scheduling problem. When the number of stages is only one, the HFS problem can be reduced to an ordinary parallel machine scheduling problem.

The processing time of the job on each stage (HFS-IPM) or each machine (HFS-UPM) is given in advance. Any machine can only handle one job at a time, and vice versa. Once machines and jobs are in working state, they are not allowed to be interrupted or preempted. A job can be processed by any machine within a processing stage. The transport time of the job or the setup time of the machine can be considered to be included in the processing time, or can be ignored. The objective of the HFS problem is to minimize the maximum completion time. The mathematical model of the HFS problem is described as follows (Eqs 1–10), with the notations and their corresponding meanings presented in Table 1. Following the rules of [66], the HFS problems studied in this work can be denoted as $HF_s \parallel C_{max}$.

$$MinC_{max} = Min\left[\underset{i \in N}{Max} \, C_{s,i}\right] \tag{1}$$

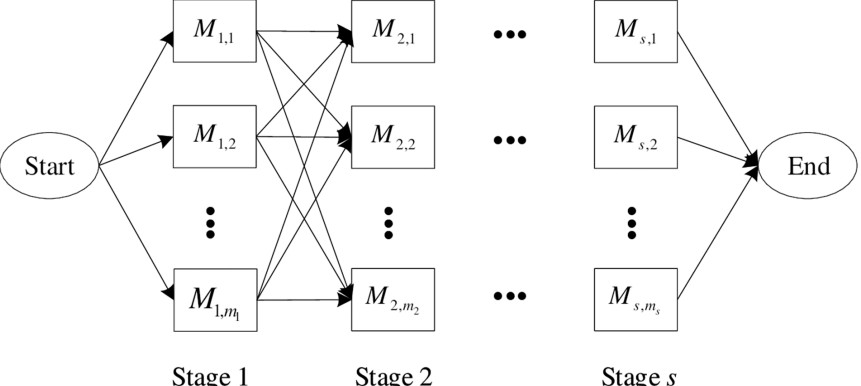

Stage 1 Stage 2 Stage $s$

**Fig 1. Layout of hybrid flow shop scheduling.**

**Table 1. Dedicated notations and the corresponding meanings.**

| Notation | Meaning |
|---|---|
| $n$ | number of jobs. |
| $N$ | set of jobs, $N = \{1, 2, \cdots, n\}$. |
| $i$ | index of the job, $i \in N$. |
| $m$ | total number of machines. |
| $s$ | total number of stages. |
| $L$ | set of stages, $L = \{1, 2, \cdots, s\}$. |
| $k$ | index for stages, $k \in L$. |
| $m_k$ | the total number of available machines at stage $k$. |
| $M_k$ | the set of available machines at stage $k$. |
| $j$ | the index of available machines at stage $k$, $j \in M_k$. |
| $B$ | a sufficiently large positive number. |
| $p_{k,i}$ | the processing time of job $i$ at stage $k$ (given in advance). |
| $C_{k,i}$ | the completion time of job $i$ at stage $k$. |
| $u_{i,i',j,k}$ | decision variable, equal to 1 if jobs $i$ and $i'$ are both assigned to machine $j$ at stage $k$, and job $i$ is the immediate predecessor of job $i'$; 0 otherwise. |
| $C_{max}$ | the maximum completion time (makespan). |

Subject to:

$$\sum_{i=0, i \neq i'}^{n} \sum_{j=1}^{m_k} u_{i,i',j,k} = 1, \ \forall i' \in N, k \in L \tag{2}$$

$$\sum_{i'=0, i' \neq i}^{n} \sum_{j=1}^{m_k} u_{i,i',j,k} = 1, \ \forall i \in N, k \in L \tag{3}$$

$$\sum_{i=0}^{n} u_{0,i,j,k} \leq 1, \ \forall k \in L, j \in M_k \tag{4}$$

$$\sum_{i=0}^{n} u_{i,0,j,k} \leq 1, \ \forall k \in L, j \in M_k \tag{5}$$

$$\sum_{i=0, i \neq i'}^{n} u_{i',i,j,k} - \sum_{i=0, i \neq i'}^{n} u_{i,i',j,k} = 0, \ \forall i' \in N, k \in L, j \in M_k \tag{6}$$

$$C_{k,i} + p_{k,i'} \cdot \sum_{j=1}^{m_k} u_{i,i',j,k} + B \cdot \left( \sum_{j=1}^{m_k} u_{i,i',j,k} - 1 \right) \leq C_{k,i'}, \ \forall k \in L, \ i, i' \in N \tag{7}$$

$$C_{k,i} \geq C_{k-1,i} + p_{k,i}, \ \forall k \in L, i \in N \tag{8}$$

$$C_{0,i} = 0, \ \forall i \in N \tag{9}$$

$$C_{s,i} \leq C_{max}, \ \forall i \in N \tag{10}$$

Constraints 2, 3 and 6 ensure that each job at every stage has exactly one immediate predecessor and one immediate successor. At the same time, they enforce that each job at every stage is processed by exactly one of the available machines. Constraints 4 and 5 ensure that, on each machine at every stage, exactly one job is processed first and exactly one job is processed last, respectively. Constraint 7 represents the time constraint between two jobs with a direct predecessor-successor relationship on the same machine at the same stage. Constraint 8

imposes the precedence constraint between different processing stages of the same job. Constraint 9 states that the earliest start time for any job is zero, indicating that there is no priority constraint on the initial processing of different jobs.

## 2.2 Probe machine

Probe machine (PM) is defined as a nine-tuple (see Fig 2)

$$PM = (X, Y, \sigma_1, \sigma_2, \tau, \lambda, \eta, Q, C), \tag{11}$$

The nine components that make up PM represent data library ($X$), probe library ($Y$), data controller ($\sigma_1$), probe controller ($\sigma_2$), probe operation ($\tau$), computing platform ($\lambda$), detector ($\eta$), true solution storage ($Q$) and residue recovery ($C$). Data library $X$ and probe library $Y$ are the two most important components of PM, and they are also the two most critical designs for PM to solve various problems. The necessary introduction to these two components is given below.

Data library $X$ can be composed of $n$ types of data, namely

$$X = \{x_1, x_2, \dots, x_n\}. \tag{12}$$

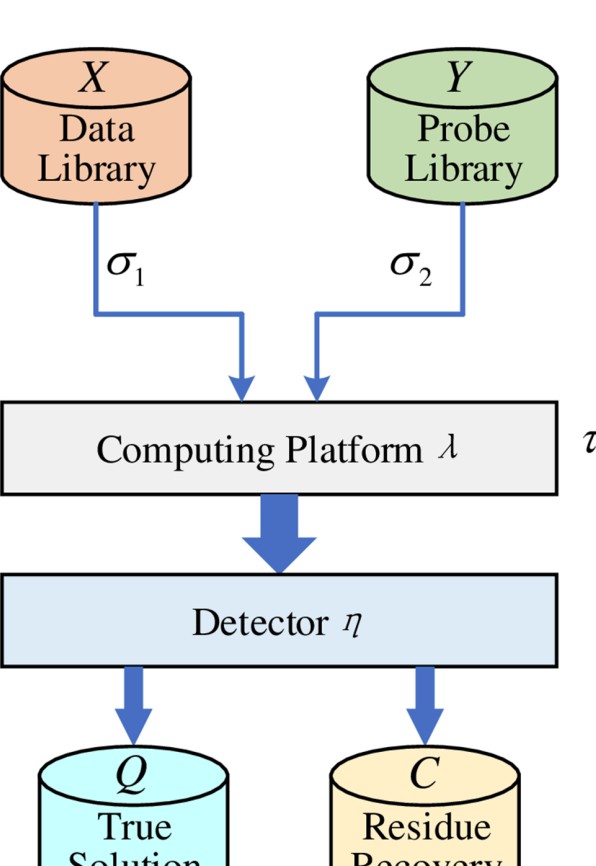

**Fig 2. Schematic diagram of PM model.**

Each type of data $x_i$ is composed of a central data body $i$ and multiple types of data fibers attached to the data body $i$, namely

$$x_i = \{i, x_i^1, x_i^2, \dots, x_i^{p_i}\}, \tag{13}$$

where the number of each data fiber may not be unique. The structural characteristics of data library $X$ well reflect the 3-D nonlinear access mode of PM.

The probe library $Y$ can be composed of multiple probe sub-libraries. Taking $Y_{it}$, one of the probe sub-libraries, as an example, it represents all possible probe sets between the two types of data $x_i$ and $x_t$. Assuming that $x_i^\alpha$ and $x_t^\beta$ are the data fibers of the two data $x_i$ and $x_t$ respectively, then the probe between $x_i^\alpha$ and $x_t^\beta$ is $\overline{x_i^\alpha x_t^\beta}$, which can be formally recorded as $\tau^{x_i^\alpha x_t^\beta} = \overline{x_i^\alpha x_t^\beta}$. Obviously, $\overline{x_i^\alpha x_t^\beta}$ is composed of complementary chains of $x_i^\alpha$ and $x_t^\beta$. According to the Watson-Crick principle, these two data fibers can be connected to realize the connection of the two data of $x_i$ and $x_t$. A test tube $Y_{it}^{\alpha\beta}$ in the probe sub-library $Y_{it}$ is also called a probe pool, which contains and only contains a large number of copies of the probe $\overline{x_i^\alpha x_t^\beta}$.

Data fibers and probes can be divided into connection and transmission according to their functional characteristics. Connective probes and connective data fibers can only achieve spatial connections between data bodies, without considering information transfer. Transitive probes and transitive data fibers can not only realize spatial connections between data bodies, but also realize the information transfer between them. In addition, a series of probe operations performed on the computing platform $\lambda$ support large-scale parallel computing. In summary, due to the advantages of nonlinear data access and complete parallel computing, the probe machine has shown strong computing power and application prospects.

## 3 The proposed improved probe machine (IPMMPO)

Although PM has advantages that TM cannot match, there is a fact that when dealing with some complex graph theory problems, only one probe operation may not be able to effectively solve them. In this section, an improved probe machine with multi-level probe operations (IPMMPO) will be proposed. In addition, HFS problem can be analyzed and solved from the perspective of disjunctive graph. Disjunctive graph is a powerful tool to transform scheduling problems into graph problems [7,77].

A simple instance of the HFS problem is given here to facilitate explanation. The instance data is shown in the Table 2, involving a total of 7 jobs, all of which have to go through three processing stages: Lathe, Planer, and Grinder. Among them, there are three available machines in the Lathe stage, labeled 1, 2, and 3; there are two available machines in the Planer stage, labeled 4, 5; There are three machines available for the Grinder stage, numbered 6, 7, and 8; the total number of machines involved is 8. The rest of the data in the table is the processing time of each job on the corresponding optional machine in each stage. It can be seen from the data in the table that this instance belongs to the unrelated parallel machine (HFS-UPM) problem.

The construction of data library and probe library is the first and most important link in the improved probe machine, and the design of probe library is based on the data library. This section first takes the simple HFS instance shown in Table 2 as an example to build its data library and probe library, and then extends the application to the general situation in the model complexity analysis part.

**Table 2**. A simple HFS instance.

| Stage | Machine | $J_1$ | $J_2$ | $J_3$ | $J_4$ | $J_5$ | $J_6$ | $J_7$ |
|---|---|---|---|---|---|---|---|---|
| Lathe | 1 | 24 | 3 | 66 | 16 | 80 | 41 | 33 |
| | 2 | 75 | 75 | 17 | 12 | 17 | 63 | 84 |
| | 3 | 51 | 17 | 71 | 91 | 22 | 87 | 21 |
| Planer | 4 | 84 | 86 | 52 | 48 | 89 | 7 | 51 |
| | 5 | 62 | 73 | 23 | 2 | 14 | 77 | 97 |
| Grinder | 6 | 39 | 99 | 33 | 14 | 38 | 56 | 63 |
| | 7 | 54 | 31 | 16 | 15 | 14 | 71 | 46 |
| | 8 | 11 | 70 | 18 | 80 | 51 | 55 | 21 |

## 3.1 Data library construction

The number of jobs in Table 2 is 7, recorded as $n = 7$, and $i$ represents the job index $(1 \leq i \leq n)$; the number of machines is 8, recorded as $m = 8$, and $j$ represents the machine index $(1 \leq j \leq m)$; the number of stages is 3, recorded as $s = 3$, and $k$ represents the stage index $(1 \leq k \leq s)$. Each job must go through three processing stages in turn, then the total number of operations of 7 jobs is $sn = 21$, and $t$ represents the operation index $(1 \leq t \leq sn)$. In this work, firstly, a data library $X_J$ about jobs is built with the operation number as the data body. The data library $X_J$, and data sub-libraries corresponding to the 7 jobs are as follows.

$$X_J = \bigcup_{i=1}^{7} X_{J_i}, \tag{14}$$

$$X_{J_1} = \{ x_1^{O_{1J},J_1,S_1}, x_2^{O_{2J},J_1,S_2}, x_3^{O_{3J},J_1,S_3} \}, \tag{15}$$

$$X_{J_2} = \{ x_4^{O_{4J},J_2,S_1}, x_5^{O_{5J},J_2,S_2}, x_6^{O_{6J},J_2,S_3} \}, \tag{16}$$

$$X_{J_3} = \{ x_7^{O_{7J},J_3,S_1}, x_8^{O_{8J},J_3,S_2}, x_9^{O_{9J},J_3,S_3} \}, \tag{17}$$

$$X_{J_4} = \{ x_{10}^{O_{10J},J_4,S_1}, x_{11}^{O_{11J},J_4,S_2}, x_{12}^{O_{12J},J_4,S_3} \}, \tag{18}$$

$$X_{J_5} = \{ x_{13}^{O_{13J},J_5,S_1}, x_{14}^{O_{14J},J_5,S_2}, x_{15}^{O_{15J},J_5,S_3} \}, \tag{19}$$

$$X_{J_6} = \{ x_{16}^{O_{16J},J_6,S_1}, x_{17}^{O_{17J},J_6,S_2}, x_{18}^{O_{18J},J_6,S_3} \}, \tag{20}$$

$$X_{J_7} = \{ x_{19}^{O_{19J},J_7,S_1}, x_{20}^{O_{20J},J_7,S_2}, x_{21}^{O_{21J},J_7,S_3} \}. \tag{21}$$

Taking $X_{J_5}$ as an example (see Eq (19)), it corresponds to the three operations of job 5. Among them, $x_{15}^{O_{15J},J_5,S_3}$ represents the third operation of job 5. The subscript 15 represents the operation number corresponding to this operation, which is designed as a data body. The superscripts $O_{15J}$, $J_5$ and $S_3$ represent the three types of data fibers connected to this data body, respectively. In order to facilitate subsequent probe operations, and then construct aggregations for feasible solutions, the number of different types of data fibers is set as follows. There is only one data fiber of the $O_{15J}$ type, which is used to realize the machine selection of the operation numbered 15; there are two data fibers of the $J_5$ type, which are used to realize the connection between the previous and subsequent operations of the same job; there is only

one data fiber of type $S_3$, which is used to mark the processing stage of the operation. Fig 3 shows the six data types contained in $X_{J_1}$ (Eq (15)) and $X_{J_5}$ (Eq (19)) respectively.

In addition, $X_J$ can also be divided into the following three data sub-libraries according to the number of stages, that is, the type of $S_k$ in the data fiber, and each sub-library contains 7 types of data.

$$X_J^{S_1} = \{x_1^{O_{1J},J_1,S_1}, x_4^{O_{4J},J_2,S_1}, x_7^{O_{7J},J_3,S_1}, x_{10}^{O_{10J},J_4,S_1},$$
$$x_{13}^{O_{13J},J_5,S_1}, x_{16}^{O_{16J},J_6,S_1}, x_{19}^{O_{19J},J_7,S_1}\}, \tag{22}$$

$$X_J^{S_2} = \{x_2^{O_{2J},J_1,S_2}, x_5^{O_{5J},J_2,S_2}, x_8^{O_{8J},J_3,S_2}, x_{11}^{O_{11J},J_4,S_2},$$
$$x_{14}^{O_{14J},J_5,S_2}, x_{17}^{O_{17J},J_6,S_2}, x_{20}^{O_{20J},J_7,S_2}\}, \tag{23}$$

$$X_J^{S_3} = \{x_3^{O_{3J},J_1,S_3}, x_6^{O_{6J},J_2,S_3}, x_9^{O_{9J},J_3,S_3}, x_{12}^{O_{12J},J_4,S_3},$$
$$x_{15}^{O_{15J},J_5,S_3}, x_{18}^{O_{18J},J_6,S_3}, x_{21}^{O_{21J},J_7,S_3}\}, \tag{24}$$

$$X_J = \bigcup_{k=1}^{3} X_J^{S_k}. \tag{25}$$

Next, build another type of data library $X_M$ based on operations and optional machines. Three data sub-libraries corresponding to three processing stages are as follows.

$$X_M^{S_1} = \{x_{11}^{O_{1M},M_1}, x_{12}^{O_{1M},M_2}, x_{13}^{O_{1M},M_3},$$
$$x_{41}^{O_{4M},M_1}, x_{42}^{O_{4M},M_2}, x_{43}^{O_{4M},M_3},$$
$$x_{71}^{O_{7M},M_1}, x_{72}^{O_{7M},M_2}, x_{73}^{O_{7M},M_3},$$
$$x_{101}^{O_{10M},M_1}, x_{102}^{O_{10M},M_2}, x_{103}^{O_{10M},M_3},$$
$$x_{131}^{O_{13M},M_1}, x_{132}^{O_{13M},M_2}, x_{133}^{O_{13M},M_3},$$
$$x_{161}^{O_{16M},M_1}, x_{162}^{O_{16M},M_2}, x_{163}^{O_{16M},M_3},$$
$$x_{191}^{O_{19M},M_1}, x_{192}^{O_{19M},M_2}, x_{193}^{O_{19M},M_3}\}, \tag{26}$$

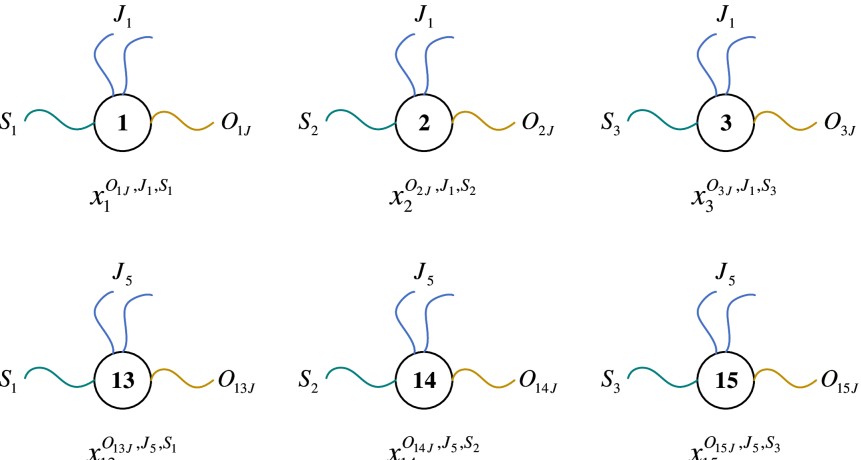

**Fig 3. Data types contained in $X_{J_1}$ and $X_{J_5}$.**

$$X_M^{S_2} = \{x_{24}^{O_{2M},M_4}, x_{25}^{O_{2M},M_5},$$
$$x_{54}^{O_{5M},M_4}, x_{55}^{O_{5M},M_5},$$
$$x_{84}^{O_{8M},M_4}, x_{85}^{O_{8M},M_5},$$
$$x_{114}^{O_{11M},M_4}, x_{115}^{O_{11M},M_5},$$
$$x_{144}^{O_{14M},M_4}, x_{145}^{O_{14M},M_5},$$
$$x_{174}^{O_{17M},M_4}, x_{175}^{O_{17M},M_5},$$
$$x_{204}^{O_{20M},M_4}, x_{205}^{O_{20M},M_5}\},$$

(27)

$$X_M^{S_3} = \{x_{36}^{O_{3M},M_6}, x_{37}^{O_{3M},M_7}, x_{38}^{O_{3M},M_8},$$
$$x_{66}^{O_{6M},M_6}, x_{67}^{O_{6M},M_7}, x_{68}^{O_{6M},M_8},$$
$$x_{96}^{O_{9M},M_6}, x_{97}^{O_{9M},M_7}, x_{98}^{O_{9M},M_8},$$
$$x_{126}^{O_{12M},M_6}, x_{127}^{O_{12M},M_7}, x_{128}^{O_{12M},M_8},$$
$$x_{156}^{O_{15M},M_6}, x_{157}^{O_{15M},M_7}, x_{158}^{O_{15M},M_8},$$
$$x_{186}^{O_{18M},M_6}, x_{187}^{O_{18M},M_7}, x_{188}^{O_{18M},M_8},$$
$$x_{216}^{O_{21M},M_6}, x_{217}^{O_{21M},M_7}, x_{218}^{O_{21M},M_8}\},$$

(28)

$$X_M = \bigcup_{k=1}^{3} X_M^{S_k}.$$

(29)

Taking $X_M^{S_1}$ as an example, it represents the data set of optional machines in the first processing stage, where each row represents a job, and different columns correspond to different optional machines. As can be seen from the data in Table 2, there are 3 machines available in the first stage (Lathe), so $X_M^{S_1}$ has 3 columns. Similarly, there are 2 optional machines in the second stage (Planer), so $X_M^{S_2}$ has 2 columns; there are 3 optional machines in the third stage (Grinder), so $X_M^{S_3}$ has 3 columns. In addition, there is a one-to-one correspondence between the data types in the data library $X_M$ and the data in Table 2, that is, there are 21 types in the first stage; 14 types in the second stage; 21 types in the third stage; a total of 56 types of data.

Specifically, for example, $x_{191}^{O_{19M},M_1}, x_{192}^{O_{19M},M_2}$ and $x_{193}^{O_{19M},M_3}$ in the last row of $X_M^{S_1}$ represent the data of three optional machines in the first stage of job 7. Taking $x_{193}^{O_{19M},M_3}$ as an example, the subscript 193 representing the data body is divided into two parts: 19 and 3. Number 3 represents optional machine 3 and corresponds to data fiber $M_3$. The number 19 in front means that the operation sequence of job 7 in the first stage is 19, which is exactly the same as that in $X_{J_7}$. Fig 4 gives a schematic of the data construction for the first and last rows in $X_M^{S_1}$.

The construction of the data library $X_J$ and $X_M$ has been completed above. The following points need to be highlighted in particular.

1. There are two fibers marked $J_i$ $(1 \le i \le n)$ in $X_J$, which are used to realize the constraint that the degree of conjunctive arc of any vertex is not larger than 2. Similarly, there are also two fibers labeled $M_j$ $(1 \le j \le m)$ in $X_M$, which will be used to realize the constraint that the degree of the disjunctive arc of any vertex does not exceed 2. Furthermore, such two fibers have the property of repelling the same kind of probes. In other words, any probe that can be matched with it can only connect to one of the two identical data fibers, and vice versa.

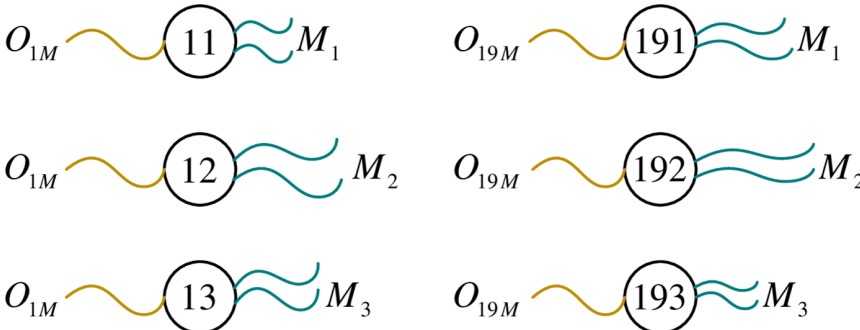

**Fig 4. Data construction of the first and last rows in $X_M^{S_1}$.**

2. The fiber marked as $S_k$ in $X_J$ is only used for data identification, and does not participate in any node connection and data transmission. Apart from this, all other fibers in $X_J$ and $X_M$ adopt transitive fibers and participate in data transfer.

3. Since there is a one-to-one mapping relationship between the data types in $X_M$ and the data in Table 2, each data body can store the unique processing time corresponding to it. In addition, the length of the data fiber marked as $M_j$ in $X_M$ is also proportional to the processing time corresponding to this data.

## 3.2 Probe library construction

Based on the above-mentioned defined data libraries, in order to further construct the acyclic feasible solution $D_S$ of the disjunctive graph $G$, it is necessary to construct three types of probe libraries to realize the following three-step probe operations respectively.

1. Firstly, realize the machine selection operation of each operation in the same processing stage.

2. Secondly, realize the connection and transmission of different operations on the same machine in the same stage.

3. Finally, for different processing stages, realize the connection and transmission between the previous and subsequent operations of the same job.

First, the probe library $Y_S$ is constructed to realize the machine selection of each operation in the same stage, and is divided into the following three sub-libraries according to the processing stage.

$$Y_S = \bigcup_{k=1}^{3} Y_{S_k}, \tag{30}$$

$$Y_{S_1} = \{\overline{O_{1J}O_{1M}}, \overline{O_{4J}O_{4M}}, \overline{O_{7J}O_{7M}}, \overline{O_{10J}O_{10M}}, \\ \overline{O_{13J}O_{13M}}, \overline{O_{16J}O_{16M}}, \overline{O_{19J}O_{19M}}\}, \tag{31}$$

$$Y_{S_2} = \{\overline{O_{2J}O_{2M}}, \overline{O_{5J}O_{5M}}, \overline{O_{8J}O_{8M}}, \overline{O_{11J}O_{11M}}, \\ \overline{O_{14J}O_{14M}}, \overline{O_{17J}O_{17M}}, \overline{O_{20J}O_{20M}}\}, \tag{32}$$

$$Y_{S_3} = \{\overline{O_{3J}O_{3M}}, \overline{O_{6J}O_{6M}}, \overline{O_{9J}O_{9M}}, \overline{O_{12J}O_{12M}},$$
$$\overline{O_{15J}O_{15M}}, \overline{O_{18J}O_{18M}}, \overline{O_{21J}O_{21M}}\}. \tag{33}$$

In the data libraries $X_J$ and $X_M$, there are a class of transitive fibers marked as $O_{tJ}$ and $O_{tM}$ ($1 \leq t \leq 21$), and these two types of fibers will be connected by specific probes in probe library $Y_S$. Taking $Y_{S_1}$ as an example, this probe sub-library will be used to implement the operation data sub-library $X_J^{S_1}$ of the first stage to perform machine selection from the corresponding candidate machine data sub-library $X_M^{S_1}$. The functions of the probe sub-libraries $Y_{S_2}$ and $Y_{S_3}$ can be deduced by analogy. Under the action of probe library $Y_S$, the data in data libraries $X_J$ and $X_M$ will form an aggregation containing two data bodies, referred to as 2-aggregation. Subsequent probe operations can continue to be executed according to any data body or any type of data fiber in the formed 2-aggregation.

Next, the probe library $Y_M$ is constructed to enable the connection of operations on the same machine within the same stage. Similarly to $Y_S$, it is divided into the following three probe sub-libraries according to the processing stage.

$$Y_M = \bigcup_{k=1}^{3} Y_M^{S_k}, \tag{34}$$

$$Y_M^{S_1} = \{\overline{x_{si-2}^{M_j} x_{sh-2}^{M_j}} \mid 1 \leq j \leq 3, s = 3, i \neq h, \ \& \ i, h \in [1,7]\}, \tag{35}$$

$$Y_M^{S_2} = \{\overline{x_{si-1}^{M_j} x_{sh-1}^{M_j}} \mid 4 \leq j \leq 5, s = 3, i \neq h, \ \& \ i, h \in [1,7]\}, \tag{36}$$

$$Y_M^{S_3} = \{\overline{x_{si}^{M_j} x_{sh}^{M_j}} \mid 6 \leq j \leq 8, s = 3, i \neq h, \ \& \ i, h \in [1,7]\}. \tag{37}$$

Take the probe $\overline{x_4^{M_1} x_7^{M_1}}$ in $Y_M^{S_1}$ as an example, where $x_4^{M_1}$ can locate and identify a 2-aggregation $A$, which contains both the data body 4 representing the operation code and the data fiber $M_1$. Specifically, this 2-aggregation $A$ is formed by aggregation of data $x_4^{O_{4J},J_2,S_1}$ in $X_J^{S_1}$ and data $x_{41}^{O_{4M},M_1}$ in $X_M^{S_1}$ via probe $\overline{O_{4J}O_{4M}}$. The subscript of $x_4^{M_1}$ corresponds to the data body 4 of $x_4^{O_{4J},J_2,S_1}$ in $A$, and the superscript $M_1$ corresponds to the data fiber $M_1$ of $x_{41}^{O_{4M},M_1}$ in $A$. Similarly, $x_7^{M_1}$ can be used to identify and locate a 2-aggregation $B$ aggregated by $x_7^{O_{7J},J_3,S_1}$ and $x_{71}^{O_{7M},M_1}$. In summary, the probe $\overline{x_4^{M_1} x_7^{M_1}}$ will connect the above two 2-aggregations $A$ and $B$ through their data fiber $M_1$, and further aggregate them into a 4-aggregation. This means that operations numbered 4 and 7 are both processed on machine $M_1$.

Finally, construct the probe library $Y_J$ to realize the connection between the previous and subsequent operations of the same job at all stages. It is divided into 7 sub-libraries according to different jobs, and the following expression can be written uniformly.

$$Y_J = \bigcup_{i=1}^{7} Y_{J_i} \tag{38}$$
$$= \{\overline{x_t^{J_i} x_{t+1}^{J_i}} \mid 1 \leq i \leq 7, s = 3, si - 2 \leq t \leq si - 1\}.$$

There are 14 types of probes in probe library $Y_J$. Take $Y_{J_2}$ as an example, $Y_{J_2} = \{\overline{x_4^{J_2} x_5^{J_2}}, \overline{x_5^{J_2} x_6^{J_2}}\}$ where $\overline{x_4^{J_2} x_5^{J_2}}$ connects the operation data $x_4^{O_{4J},J_2,S_1}$ and $x_5^{O_{5J},J_2,S_2}$ belonging to the same job $J_2$ but

at different stages through its transitive data fiber $J_2$, so as to realize the sequence constraints between the previous and subsequent operations.

**Library design principles:**

1. **Isomorphism:** The construction of data libraries is the first and most important. $X_J$ and $X_M$ strictly mirror the binary structures of the conjunctive graph and disjunctive graph, respectively.

2. **Completeness:** For any instance of size $(n,m,s)$, $|X_J| = n \cdot s$, $|X_M| = \sum_{k=1}^{s} n \cdot m_k$ (see Sect 3.4 Theorem 1 for details).

3. **Probe-targeted:** The construction of probe libraries is completely dependent on data libraries. Each probe library addresses a specific scheduling sub-decision.

## 3.3 Structure and computational framework of IPMMPO

After the data library and probe library for the HFS problem are constructed, this section will give the structure and computational framework of IPMMPO (see Fig 5). The specific computation steps are as follows.

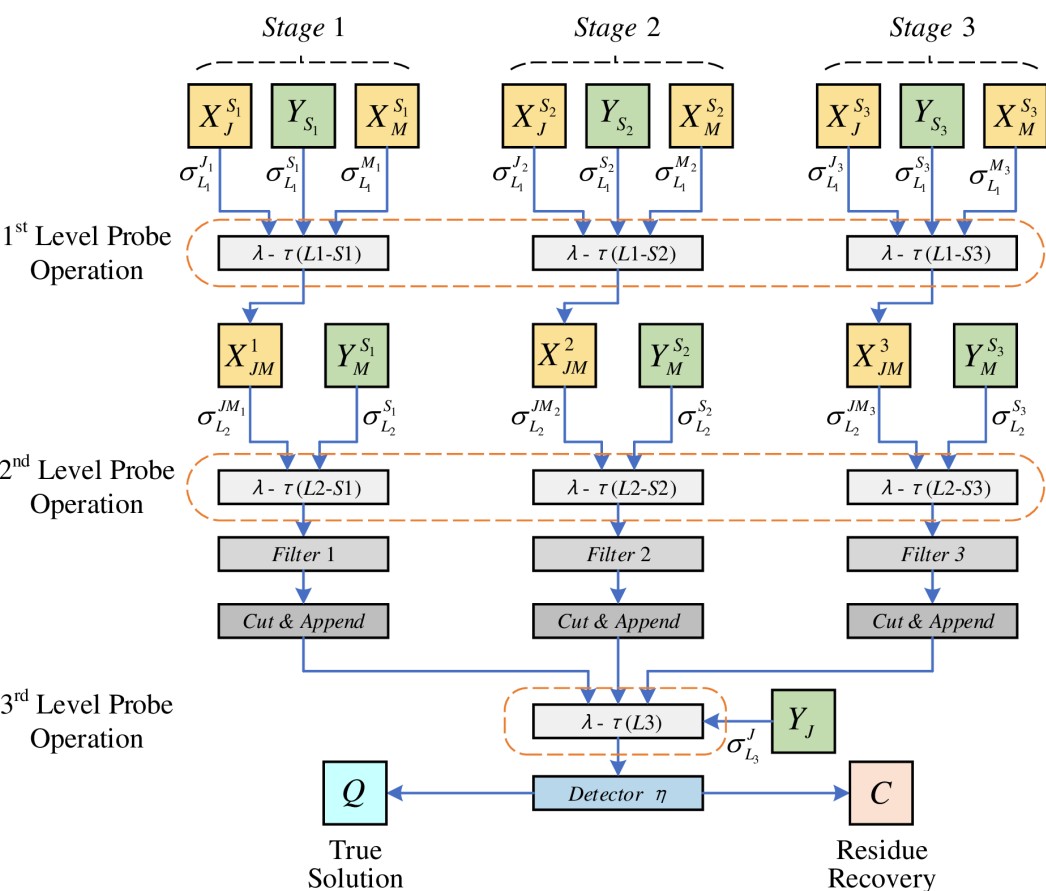

**Fig 5. Structure and computational framework of IPMMPO.**

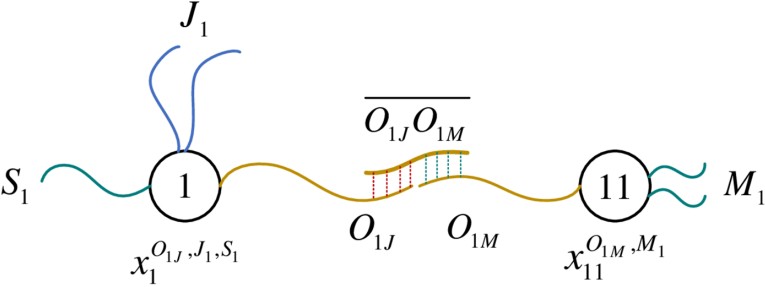

**Fig 6. Probe $\overline{O_{1J}O_{1M}}$ combines $x_1^{O_{1J},J_1,S_1}$ and $x_{11}^{O_{1M},M_1}$ as a 2-aggregation.**

1. **First level probe operation.** Through three parallel probe operations, the random selection of machines for each operation in the three stages is realized respectively. Under the action of $Y_{S_k}$, the operation data sub-library $X_J^{S_k}$ and the machine data sub-library $X_M^{S_k}$ perform the first-level probe operation, $1 \leq k \leq 3$. Specifically, each data in $X_M^{S_k}$ and $X_J^{S_k}$ has a large number of copies, and the number of copies of the data in $X_J^{S_k}$ must be an integer multiple of the number of optional machines in this stage to ensure the randomness of machine selection for operations. Fig 6 gives an example of a 2-aggregation that may occur after $X_J^{S_1}$ and $X_M^{S_1}$ perform the first-level probe operation under the action of $Y_{S_1}$, which means that the first stage of job 1 chooses to be processed on machine 1.

2. **Second level probe operation.** After executing the above probes, all possible 2-aggregation sets formed in each stage are denoted as $X_{JM}^k$ ($1 \leq k \leq 3$). Introduce the probe library $Y_M$ to perform the second-level probe operation. Under the action of $Y_M^{S_k}$, operations in $X_{JM}^k$ that are processed on the same machine will be chained in random order. Since the number of $M_j$-type data fibers in $X_M^{S_k}$ is 2, it can ensure that the degree of each operation node in the generated feasible solution is at most 2. According to the characteristics of the data fiber expressed at the end of Sect 3.1, after the second-level probe operation, it is impossible to form a ring similar to "7-13-7", but it is possible to form a ring similar to "7-13-10-7". The operation sequences generated in the three stages exist in the form of rings, and each ring structure corresponds to a machine.

3. **Detect and filter operations.** Probes that do not participate in the probe operation are filtered out. Detect the generated multi-aggregation, and filter out the multi-aggregation that contains more than

$$\{1 + \lceil n/\min(M_k)\rceil\} \times 2 \tag{39}$$

data bodies, that is, the number of 2-aggregation exceeds

$$1 + \lceil n/\min(M_k)\rceil, \tag{40}$$

so as to achieve machine load balancing. Here $M_k$ represents the number of available machines in stage $k$.

4. **Cut and mark operations.** After the operation of Step 3, the rest are valid ring structures. The cutting operation is carried out at the longest connection point of the data fiber M in the ring structure, and the starting and ending point markings are attached while the original probe is removed. For example, for three-stage multi-aggregation,

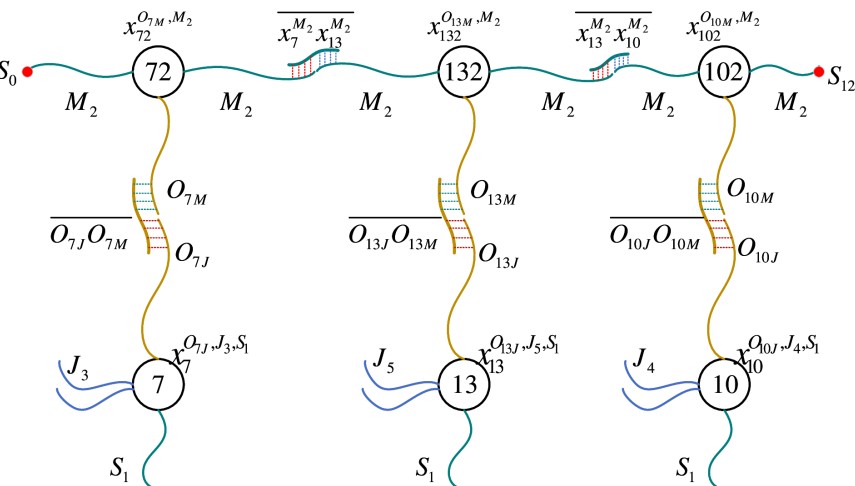

**Fig 7. Schematic diagram of the cut-and-mark operation.**

their start and end points are marked as $S_0$ and $S_{12}$, $S_{21}$ and $S_{23}$, and $S_{32}$ and $S_*$, respectively. Fig 7 shows the schematic diagram of the ring structure "7-13-10-7" after cutting-marking operation. This means that in the first stage, operations numbered 7, 13, and 10 are all on machine $M_2$ and processed in the order 7-13-10.

5. **Third level probe operation.** This operation is responsible for connecting the operations of each stage. Consider the following two points: (1) There is no intersection between machines in different stages and they are independent of each other; (2) However, the same job has operational priority constraints between different stages, that is, the successive operations of the same job closely link different stages. Therefore, the probe library $Y_J$ is introduced, the third-level probe operation is performed based on the data fiber $J_i$ ($1 \leq i \leq 7$), and a set of conjunctive arcs related to the operation is established, and then the three stages are associated. Apparently, no ring structure was generated after the third-level probe operation.

6. **Detect and identify operations.** Assume that the length between any two operations $t$ and $h$ ($t \neq h$, & $1 \leq t, h \leq 21$) is denoted as $L(t,h)$. Then, the $L(S_0, S_*)$ value of the finally generated multi-aggregation is detected by transitive fibers and probes, which is regarded as the feasible solution corresponding to the multi-aggregation. $S_0$ and $S_*$ here correspond to the virtual start and end points 0 and * in the disjunctive graph model, respectively. Finally, the multi-aggregation with the smallest $L(S_0, S_*)$ value is filtered into the true solution storage $Q$, which is the optimal solution of the problem.

Fig 8 shows an optimal solution for this instance. Specifically, the colored disjunctive arcs between operation nodes coded by numbers are realized through Steps 2 to 4 above, while the black conjunctive arcs are realized through Step 5. The nodes in Fig 8 have 7 rows and 3 columns. Each row represents three sequential operations of the same job from left to right. Each column then corresponds to a processing stage. Acyclic structures marked with the same color in each column correspond to processing sequences on the same machine at that stage. Correspondingly, Fig 9 shows the scheduling Gantt chart corresponding to the optimal solution.

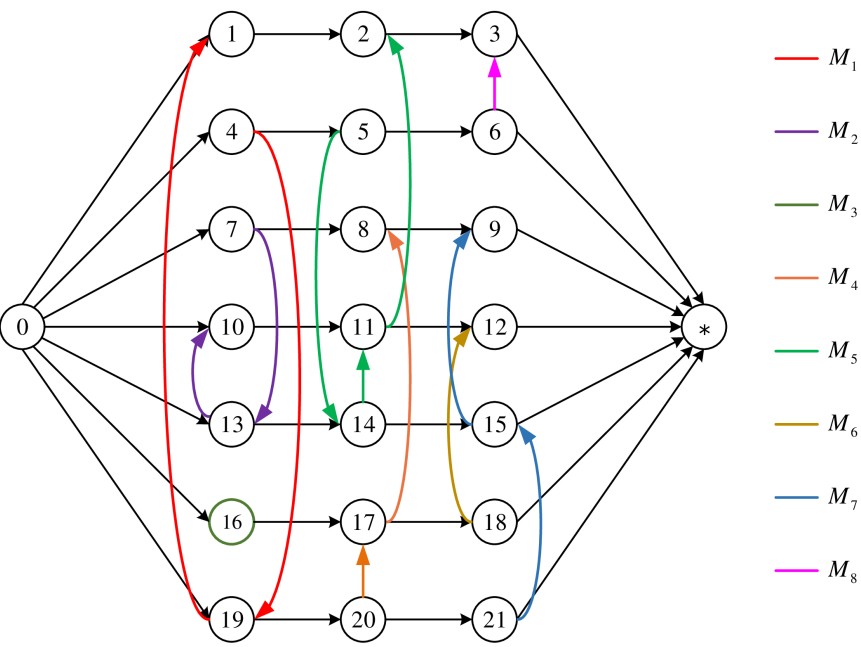

**Fig 8. An optimal solution for the HFS instance.**

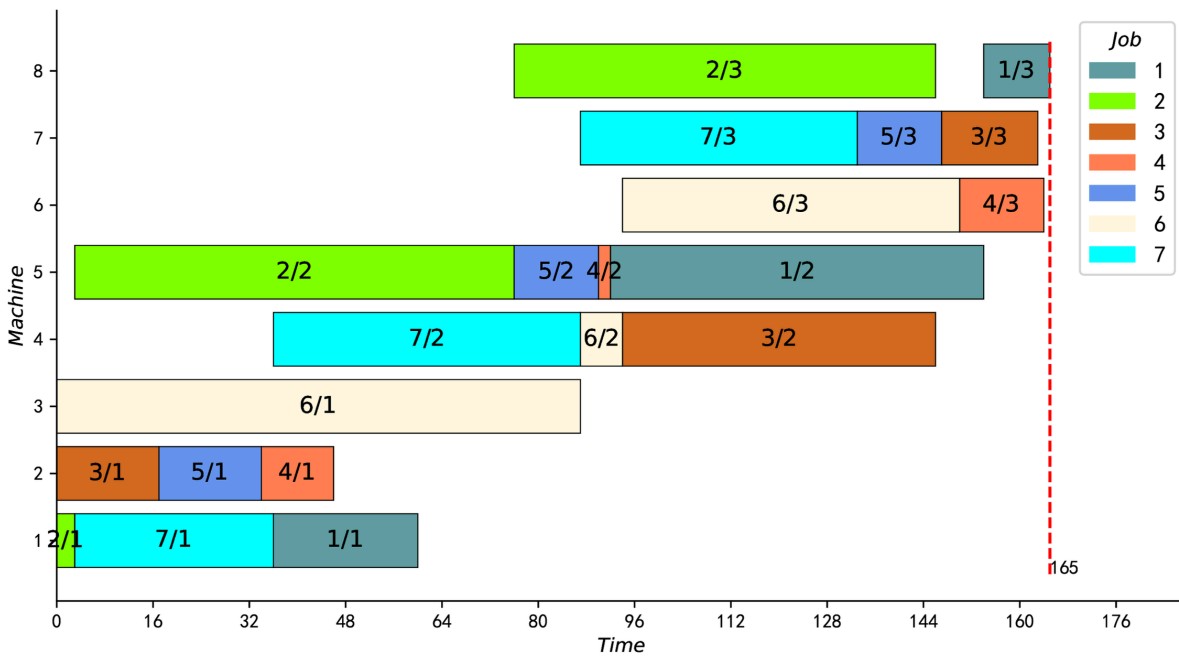

**Fig 9. The scheduling Gantt chart.**

## 3.4 Operational complexity analysis of IPMMPO

The essence of the probe machine lies in DNA computation, and its time complexity arises from the six computational operations introduced in Sect 3.3. Since all the above operations

can be completed in the laboratory through a constant number of biological experiments, it is reasonable to assume that the time complexity of these operations is $O(1)$ [75]. In previous studies [44,61,75,76], the same method has been used to analyze the time complexity of DNA computation, so this study also adopts the same approach. The following content will focus on space complexity.

Compared with jobs, machines are relatively scarce resources, meaning that the number of machines $m$ is typically much smaller than the number of jobs $n$. This is also a premise for studying various scheduling problems. Additionally, the number of available machines in each stage is at least 1, which means that the number of machines $m$ is always greater than the number of stages $s$. In typical scenarios, $s$ is generally considered to be a constant much smaller than $n$. Therefore, in the worst-case scenario, we may consider $n$ as an upper bound for both $m$ and $s$.

**Theorem 1.** *Without loss of generality, $n \geq m \geq s$ is assumed. The total number of data types in two data libraries is bounded by $O(n^2)$.*

**Proof 1.** The data types in the data library $X_J$ correspond to the total operations of the HFS problem one-to-one, and the total number of operations $n \cdot s$ are jointly determined by $n$ and $s$. So the total number of data types in $X_J$ is bounded by $O(n^2)$. The total number of data types in data libraries $X_M$, denoted as $n \cdot m$, can be abbreviated as $n^2$. So the total number of data types in $X_M$ is bounded by $O(n^2)$. Therefore, the total number of data types in data libraries $X_J$ and $X_M$ is bounded by $O(n^2) + O(n^2) = O(n^2)$.

**Theorem 2.** *Without loss of generality, $n \geq m \geq s$ is assumed. The total number of probe types in the three probe libraries is bounded by $O(n^3)$.*

**Proof 2.** The probe types in the probe library $Y_S$ are determined by $n$ and $s$, and the total number is $n \cdot s$, so the total number in $Y_S$ is bounded by $O(n^2)$. The total number of types in the probe library $Y_M$ is $m \times C_n^2 = mn(n-1)/2$, so it is bounded by $O(n^3)$. Finally, the total number of types in $Y_J$ is $s$, so it is bounded by $O(n)$. Therefore, the total number of probe types in the three probe libraries is bounded by $O(n^2) + O(n^3) + O(n) = O(n^3)$.

To sum up, the space complexity is bounded by $O(n^3)$.

## 4 Constraint programming simulation modeling based on IPMMPO (IPMMPO-CP)

### 4.1 Data preprocessing based on IPMMPO

This section preprocesses the HFS instance data completely according to data libraries constructed in IPMMPO. Corresponding to the two databases $X_J$ and $X_M$ designed in Sect 3.1, the tuple set $P_{OId,i,k}$ based on 'operation-job' and the tuple set $M_{OId,j,pt}$ based on 'operation-optional machine' are respectively constructed here.

$P_{OId,i,k}$ is a set of tuples corresponding to $X_J$, in the form of $\{< t,i,k >| 1 \leq t \leq sn, 1 \leq i \leq n, 1 \leq k \leq s\}$, where the operation number $t$ corresponds to the data body of $X_J$, and the job number $i$ corresponds to the data fiber $J_i$ of $X_J$. The stage $k$ where $t$ is located corresponds to the data fiber $S_k$ of $X_J$. Taking the three data in $X_{J_1}$ shown in Fig 3 as an example, $x_1^{O_{1J},J_1,S_1}$, $x_2^{O_{2J},J_1,S_2}$, and $x_3^{O_{3J},J_1,S_3}$ correspond to the three tuples of <1,1,1>, <2,1,2>, and <3,1,3> in $P_{OId,i,k}$, representing the three operations of job 1. Similarly, the three data $x_{13}^{O_{13J},J_5,S_1}$, $x_{14}^{O_{14J},J_5,S_2}$, and $x_{15}^{O_{15J},J_5,S_3}$ in $X_{J_5}$ correspond to the three tuple sets of <13,5,1>, <14,5,2>, and <15,5,3> in $P_{OId,i,k}$, representing the three operations of job 5, and so on. The data in $X_J$ is in a one-to-one mapping relationship with the tuples in $P_{OId,i,k}$.

$M_{OId,j,pt}$ is a set of tuples corresponding to $X_M$, in the form of $\{< t,j,pt >| 1 \leq t \leq sn, 1 \leq j \leq m\}$, where $pt$ represents the processing time given in advance, depending on the specific instance. $t$ and $j$ correspond to operation codes and optional machines in $X_M$, respectively.

Taking the data in $X_M^{S_1}$ shown in Fig 4 as an example, $x_{11}^{O_{1M},M_1}$, $x_{12}^{O_{1M},M_2}$, and $x_{13}^{O_{1M},M_3}$ correspond to the three tuple sets of <1,1,24>, <1,2,75>, and <1,3,51> in $M_{OId,j,pt}$, respectively. Similarly, the three data $x_{191}^{O_{19M},M_1}$, $x_{192}^{O_{19M},M_2}$, and $x_{193}^{O_{19M},M_3}$ correspond to the three tuple sets of <19,1,33>, <19,2,84>, and <19,3,21> in $M_{OId,j,pt}$, and so on. The data in $X_M$ is also in a one-to-one mapping relationship with the tuples in $M_{OId,j,pt}$.

The tuple sets $P_{OId,i,k}$ and $M_{OId,j,pt}$ are completely consistent with data libraries $X_J$ and $X_M$ not only in their tuple content but also in the number of tuples they contain. That is, the number of tuples in $P_{OId,i,k}$ and $M_{OId,j,pt}$ are also $ns$ and $nm$, respectively. $P_{OId,i,k}$ and $M_{OId,j,pt}$ can be related by a common operation index $OId$.

Data preprocessing contributes to enhancing the interpretability of data, thereby providing a more reliable and effective foundation for subsequent modeling process. The data structure $M_{OId,j,pt}$ proposed in this work can be effectively compatible with both HFS-IPM and HFS-UPM data types. The tuples mentioned above, <1,1,24>, <1,2,75>, and <1,3,51>, serve as examples. They indicate that for operation 1, the optional machines are 1, 2, and 3, with corresponding processing times of 24, 75, and 51 respectively. It is evident that the processing time of the same operation on different optional machines is not the same, which constitutes an HFS-UPM problem. For the HFS-IPM problem, the possible tuples are in the form of <1,1,24>, <1,2,24> and <1,3,24>. That is, the processing time of the same operation on different optional machines is the same. In summary, the data structure $M_{OId,j,pt}$ can satisfy the data feeding of these two types of HFS at the same time without affecting the construction of the model.

## 4.2 The proposed IPMMPO-CP

CP method can not only obtain high-quality feasible solutions for large-scale problems in a reasonable time, but also prove that the best feasible solution found is optimal when the search automatically terminates [59,64]. The two types of decision variables that may be involved in scheduling problems are interval variables and interval sequence variables. The three attributes start, end and length of the interval variable correspond to the start time, end time and size of a time interval, respectively. Based on the tuple set $P_{OId,i,k}$, this work defines an interval variable *PRO* to represent the time interval associated with all jobs and processing stages that must appear in the final solution. Based on the tuple set $M_{OId,j,pt}$, an optional interval variable *Itvs* is defined to represent an optional machine-dependent time interval that does not necessarily appear in the final solution.

**Parallel machine scenario adaptability:** The unified data structure $M_{OId,j,pt}$ inherently supports both HFS-IPM and HFS-UPM scenarios. For HFS-IPM, processing times satisfy $\forall j_1, j_2 : pt(t, j_1) = pt(t, j_2)$; for HFS-UPM, $pt(t,j)$ varies arbitrarily. This design eliminates scenario-specific modeling.

**Scenario switching mechanism:** The transition between *standard* and *no-wait* scenarios requires only one constraint change: Replace `endBeforeStart` (Eq 44) with `endAtStart` (Eq 48), demonstrating extreme flexibility in multi-scenario adaptation.

Further, based on the interval variable *Itvs* above, an interval sequence variable *Seq* is defined to represent the possible ordering of this interval variable. The dedicated notations involved in the IPMMPO-CP model and their meanings are shown in Table 3.

After the interval variables and interval sequence variables are properly defined, the functions based on these decision variables can further express constraints on the scheduling problem, which makes the constraint expression of CP modeling both concise and rich. The proposed IPMMPO-CP will be explained in detail below. Eq (41) gives the objective function for the HFS problem. The function *endOf*(.) is used here, which is an integer expression based

**Table 3. Dedicated notations and the corresponding meanings.**

| Notation | Meaning |
|---|---|
| $M$ | Set of all machines, $M = \{1, 2, \cdots, m\}$. |
| $OId$ | An encoded index of all operations for all jobs, $OId \in \{1, 2, \cdots, sn\}$. |
| $P_{OId,i,h}$ | A set of triplets that stores the operation index $OId$ corresponding to job $i$ in processing stage $h$. |
| $M_{OId,j,pt}$ | A set of triplets that stores the optional machines and their processing time corresponding to the $OId$ in $P_{OId,i,h}$. |
| $MchQty$ | A known vector storing the number of available machines in each stage. |
| $PRO[o]$ | An interval variable associated with $o \in P_{OId,i,h}$. |
| $Itvs[p]$ | An optional interval variable associated with $p \in M_{OId,j,pt}$. |
| $Seq[k]$ | An interval sequence variable, which is a set of orderings for the interval variable $Itvs$, where $k \in M$. |

on an interval variable to return the end time of the last operation of all jobs.

$$Minimize \max_{i} \{endOf(PRO[o])\}$$
$$\forall i \in N, o \in P_{OId,i,h} : o.h = s \tag{41}$$

Subject to:

$$cumulFunction\ f(k) = \sum_{o.OId=1}^{sn} pulse(PRO[o], 1)$$
$$\forall k \in L, o \in P_{OId,i,h} : o.h = k \tag{42}$$

Eq (42) first defines the cumulative function of the machine occupied in each processing stage, where *cumulFunction* is the cumulative function, which is constructed by the algebraic sum of the atomic function *pulse*(.). Fig 10 shows the schematic diagram of this atomic function, where *PRO* is the defined interval variable, and *e* represents the number of resources (i.e., machines) occupied by the interval variable.

In short, in stage *k*, as long as there is a machine allocated and not occupied repeatedly, its pulse value is recorded as 1. The cumulative function $f(k)$ only counts the total number of

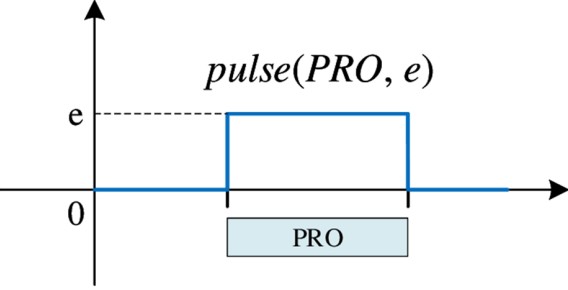

**Fig 10. Schematic diagram of the atomic function *pulse*(.).**

different parallel machines occupied in stage $k$.

$$f(k) \leq MchQty(k), \forall k \in L \tag{43}$$

Eq (43) uses the cumulative function defined in Eq (42) to impose specific constraints on the available machine capacity in each stage. In other words, Eqs (42), (43) are a set of capacity constraints for a limited total number of machines available over each processing stage.

$$endBeforeStart\ (PRO[o_1], PRO[o_2]),$$
$$\forall i \in N, o_1, o_2 \in P_{OId,i,h} : o_1.i = o_2.i\ \&\ 1 + o_1.h = o_2.h \tag{44}$$

Eq (44) implements the precedence constraint over the sequential operation order of the same job, where the function $endBeforeStart(.)$ is a temporal constraint based on interval variables.

$$alternative(PRO[o], Itvs[p]),$$
$$\forall o \in P_{OId,i,h}, \forall p \in M_{OId,j,pt} : o.OId = p.OId \tag{45}$$

Eq (45) is an alternative constraint to ensure that each operation is processed on only one of its corresponding available machines. The function $alternative(.)$ used here is a alternative constraint over interval variables, specifically for modeling discrete selections in scheduling.

$$noOverlap(Seq[k]), \forall k \in M \tag{46}$$

Eq (46) is a non-overlapping constraint based on interval sequence variables, which means that, on any machine in any stage, any two operations cannot overlap in the time domain.

## 5 The experimental comparison based on HFS-IPM instances

In order to fully verify the performance of IPMMPO-CP, this section first conducts comparative experiments based on a large number of HFS-IPM instances. This section involves a total of 168 instances of different scales, and 9 representative or state-of-the-art algorithms. The hardware environment of the experiments is: Intel Core i5, 2.9GHz, and the running memory is 16GB.

### 5.1 Experiment 1: Small and medium scale

This subsection first selects 18 difficult instances (see Table 4) from the benchmark published by [79]. Take "j15c5d2" as an example to explain this benchmark, where the letters "*j*" and "*c*" represent "job" and "stage" respectively. And "*d*" represents the type of available machine distribution in all stages, which is one of the four types *a*, *b*, *c*, and *d*. It should be noted that in both types "*a*" and "*b*", there is such a stage, in which only one machine is available. That is, in a stage where there is only one available machine, there is no need to consider the problem of machine allocation, so instances with types *a*, *b* are easier to solve. Whereas for types "*c*" and "*d*", there are more than two machines available at any stage, that is, machine allocation needs to be considered for each stage. So we use instances with types "*c*" and "*d*" for comparison and discussion. The comparison algorithms involved are hybrid evolutionary algorithm (HEA, [7]), discrete artificial bee colony (DABC, [21]), particle swarm optimization (PSO, [4]) and artificial immune system (AIS, [78]).

The last two columns of Table 4 present the computation results ($C_{max}$) of our IPMMPO-CP and the CPU time used (in seconds). Bold data with an asterisk (*) in the "$C_{max}$" column

**Table 4.** Comparison based on hard instances from [79].

| Instances | HEA | DABC | PSO | AIS | our CP | |
|---|---|---|---|---|---|---|
| | | | | | $C_{max}$ | CPU(s) |
| j10c10c1 | 114 | 114 | 115 | 115 | 114 | **1.1!** |
| j10c10c2 | 116 | 116 | 117 | 119 | 116 | **1.7!** |
| j10c10c3 | 116 | 116 | 116 | 116 | 116 | **1.8!** |
| j10c10c4 | 119 | 120 | 120 | 120 | 119 | **4.2!** |
| j10c10c5 | 125 | 125 | 125 | 126 | 125 | **3.6!** |
| j10c10c6 | 105 | 105 | 106 | 106 | **104**\* | **3.4!** |
| j15c5c1 | 85 | 85 | 85 | 85 | 85 | **1.9!** |
| j15c5c2 | 90 | 90 | 90 | 91 | 90 | **1.1!** |
| j15c5c3 | 87 | 87 | 87 | 87 | 87 | **2!** |
| j15c5c4 | 89 | 89 | 89 | 89 | 89 | **1.8!** |
| j15c5c5 | 74 | 74 | 74 | 74 | 74 | **4.3!** |
| j15c5c6 | 91 | 91 | 91 | 91 | 91 | **0.68!** |
| j15c5d1 | 167 | 167 | 167 | 167 | 167 | **0.3!** |
| j15c5d2 | 84 | 84 | 84 | 84 | 84 | **4.4!** |
| j15c5d3 | 82 | 82 | 82 | 83 | 82 | **4.8!** |
| j15c5d4 | 84 | 84 | 84 | 84 | 84 | **4.5!** |
| j15c5d5 | 79 | 79 | 79 | 80 | 79 | **3.5!** |
| j15c5d6 | 81 | 81 | 81 | 82 | 81 | **4.9!** |

indicates that the current best feasible solution for the corresponding instance was improved. The data in bold with an exclamation mark (!) in the last column means that the algorithm automatically stopped at that time, which means that the best feasible solution found is proved to be optimal. It is clear from Table 4 that the solution for all instances can be automatically stopped within 5 seconds. Therefore, the current best feasible solutions for these 18 hard instances are also proved to be optimal for the first time. Fig 11 shows the optimal Gantt chart of instance j10c10c6.

We next select another 110 small and medium-sized instances from the newly released benchmark instances [65] for test comparison. The comparison algorithms involved in this round of experiments are HEA [7], simulated annealing (SA) [28], and chaos-enchanced simulated annealing (CSA) [28]. In addition, the upper bound (UB) is the result obtained by the publisher of this series of benchmark instances according to the improved iterative greedy algorithm (IG) described by [68]. The detailed experimental results and comparisons are presented in Table 5.

The last two columns of Table 5 present the computation results ($C_{max}$) of our IPMMPO-CP and the CPU time used (in seconds). Among them, "$C_{max}$" is the best feasible solution found by IPMMPO-CP. As mentioned earlier, the data marked with an exclamation mark (!) in the last column indicates that IPMMPO-CP automatically stops searching at that moment, which means that the optimal solution for the corresponding instance is found. Solving for all instances either stops automatically because an optimal solution is found, or terminates when the time limit (900s) is reached. Take "20_15_9" as an example, the first number "20" means the number of jobs is 20, the second number "15" represents a total of 15 processing stages, and the last number "9" represents the serial number of the series of instances.

The following conclusions can be drawn from Table 5.

- First, the best feasible solution for 60 of the 110 instances is updated, and furthermore, the best feasible solution for only one instance (20_20_3) is not as good as that found by HEA, but better than the results of the other three comparison algorithms;

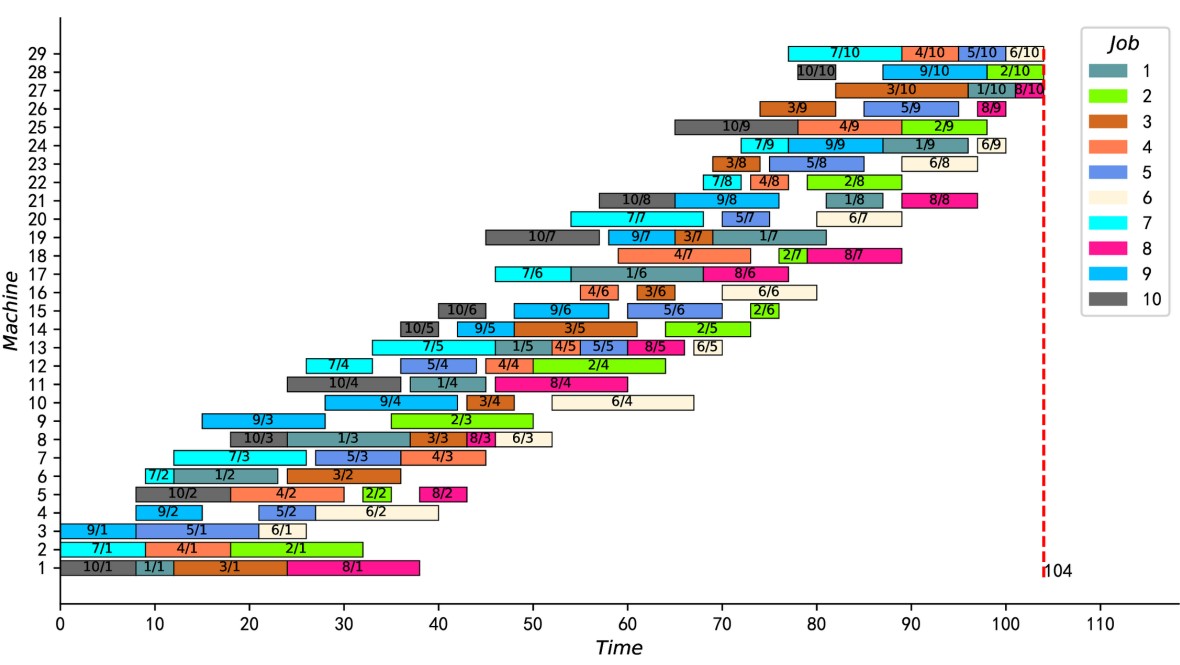

**Fig 11. Scheduling Gantt chart for instance j10c10c6.**

- Second, from the marked data in the last column, the current best feasible solutions for 83 of the 110 instances are proved to be optimal for the first time;
- Third, as with all heuristics, the computation time will inevitably increase as the problem size increases, but it is well within acceptable and reasonable limits.

**Table 5. Comparison of small- and medium-sized instances from [65].**

| Instances | UB | CSA | SA | HEA | our CP | |
|---|---|---|---|---|---|---|
| | | | | | $C_{max}$ | CPU(s) |
| 10_10_1 | 806 | 794 | 794 | 794 | 794 | **2.7!** |
| 10_10_2 | 785 | 785 | 785 | 785 | 785 | **2.4!** |
| 10_10_3 | 755 | 755 | 755 | 755 | 755 | **1.1!** |
| 10_10_4 | 922 | 914 | 914 | 914 | 914 | **1.4!** |
| 10_10_5 | 969 | 957 | 957 | 957 | 957 | **1.5!** |
| 10_10_6 | 1001 | 1001 | 1001 | 1001 | 1001 | **1.1!** |
| 10_10_7 | 947 | 942 | 940 | 938 | 938 | **2.7!** |
| 10_10_8 | 545 | 543 | 543 | 543 | 543 | **1.4!** |
| 10_10_9 | 516 | 510 | 510 | 510 | 510 | **2.4!** |
| 10_10_10 | 684 | 684 | 684 | 684 | 684 | **5.4!** |
| 10_15_1 | 959 | 959 | 959 | 959 | 959 | **1.3!** |
| 10_15_2 | 1290 | 1290 | 1290 | 1290 | 1290 | **2.1!** |
| 10_15_3 | 1091 | 1091 | 1091 | 1091 | 1091 | **1.3!** |
| 10_15_4 | 875 | 866 | 866 | 866 | 866 | **3.1!** |
| 10_15_5 | 883 | 882 | 882 | 882 | **876**\* | **20.5!** |
| 10_15_6 | 843 | 836 | 836 | 836 | 836 | **1.1!** |
| 10_15_7 | 912 | 903 | 903 | 901 | **900**\* | **21.5!** |
| 10_15_8 | 770 | 765 | 765 | 765 | 765 | **8.5!** |
| 10_15_9 | 764 | 751 | 751 | 755 | 751 | **2.1!** |
| 10_15_10 | 866 | 847 | 848 | 847 | 847 | **7.9!** |

(*Continued*)

Table 5. (Continued)

| Instances | UB | CSA | SA | HEA | our CP | |
|---|---|---|---|---|---|---|
| | | | | | $C_{max}$ | CPU(s) |
| 10_20_1 | 1353 | 1353 | 1353 | 1347 | **1345*** | **1.8!** |
| 10_20_2 | 1156 | 1155 | 1155 | 1155 | 1155 | **8.5!** |
| 10_20_3 | 1503 | 1503 | 1503 | 1498 | 1498 | **12.5!** |
| 10_20_4 | 1483 | 1463 | 1463 | 1452 | 1452 | **14.2!** |
| 10_20_5 | 1505 | 1482 | 1482 | 1482 | **1475*** | **17.9!** |
| 10_20_6 | 1309 | 1306 | 1294 | 1302 | 1294 | **10.1!** |
| 10_20_7 | 1420 | 1410 | 1414 | 1410 | 1410 | **8.2!** |
| 10_20_8 | 1522 | 1517 | 1517 | 1517 | 1517 | **16.5!** |
| 10_20_9 | 902 | 883 | 887 | 887 | 883 | **3.9!** |
| 10_20_10 | 1099 | 1088 | 1089 | 1088 | **1087*** | **14!** |
| 15_5_1 | 486 | 480 | 480 | 483 | 480 | **10!** |
| 15_5_2 | 423 | 421 | 422 | 423 | **420*** | **80!** |
| 15_5_3 | 504 | 503 | 503 | 503 | 503 | **3.9!** |
| 15_5_4 | 440 | 436 | 436 | 439 | 436 | **13!** |
| 15_5_5 | 420 | 417 | 417 | 417 | **416*** | **2.4!** |
| 15_5_6 | 414 | 411 | 411 | 413 | 411 | **12.5!** |
| 15_5_7 | 484 | 484 | 484 | 484 | **483*** | **12!** |
| 15_5_8 | 525 | 520 | 520 | 524 | 520 | **8.9!** |
| 15_5_9 | 557 | 548 | 548 | 554 | **547*** | **7.7!** |
| 15_5_10 | 443 | 442 | 440 | 443 | 440 | **2.1!** |
| 15_10_1 | 757 | 754 | 752 | 752 | **748*** | **278!** |
| 15_10_2 | 704 | 697 | 697 | 694 | **691*** | **53!** |
| 15_10_3 | 853 | 853 | 853 | 853 | 853 | **15!** |
| 15_10_4 | 886 | 879 | 880 | 883 | **875*** | **150!** |
| 15_10_5 | 1087 | 1076 | 1076 | 1084 | 1076 | **11!** |
| 15_10_6 | 1042 | 1042 | 1042 | 1042 | **1038*** | **12!** |
| 15_10_7 | 1020 | 1020 | 1020 | 1020 | 1020 | **1.6!** |
| 15_10_8 | 1011 | 1011 | 1011 | 1011 | 1011 | **7.9!** |
| 15_10_9 | 659 | 643 | 646 | 651 | **640*** | **375!** |
| 15_10_10 | 736 | 733 | 734 | 734 | **730*** | **170!** |
| 15_15_1 | 1029 | 1023 | 1025 | 1027 | **1022*** | **339!** |
| 15_15_2 | 1059 | 1055 | 1056 | 1056 | **1053*** | 378 |
| 15_15_3 | 1151 | 1144 | 1144 | 1144 | 1144 | **28!** |
| 15_15_4 | 1173 | 1160 | 1162 | 1167 | **1158*** | 18.5 |
| 15_15_5 | 1190 | 1183 | 1182 | 1184 | **1181*** | **239!** |
| 15_15_6 | 1168 | 1163 | 1164 | 1163 | **1159*** | **14.5!** |
| 15_15_7 | 1570 | 1549 | 1557 | 1553 | **1537*** | **101!** |
| 15_15_8 | 943 | 938 | 940 | 942 | **932*** | **165!** |
| 15_15_9 | 909 | 899 | 902 | 898 | **891*** | **295!** |
| 15_15_10 | 876 | 871 | 871 | 876 | **867*** | **527!** |
| 15_20_1 | 1264 | 1255 | 1258 | 1260 | **1251*** | **245!** |
| 15_20_2 | 1564 | 1564 | 1564 | 1564 | 1564 | **4.5!** |
| 15_20_3 | 1213 | 1208 | 1208 | 1206 | **1202*** | **198!** |
| 15_20_4 | 1557 | 1557 | 1557 | 1557 | 1557 | **9.8!** |
| 15_20_5 | 1558 | 1537 | 1536 | 1553 | **1535*** | **25!** |
| 15_20_6 | 1692 | 1686 | 1686 | 1675 | **1669*** | **11.5!** |
| 15_20_7 | 1731 | 1699 | 1699 | 1693 | **1682*** | **125!** |
| 15_20_8 | 1712 | 1709 | 1707 | 1699 | **1692*** | **25!** |
| 15_20_9 | 1003 | 990 | 993 | 990 | **981*** | **85!** |
| 15_20_10 | 1098 | 1085 | 1087 | 1092 | **1083*** | 131 |
| 20_5_1 | 660 | 660 | 660 | 660 | 660 | **3.8!** |
| 20_5_2 | 587 | 585 | 585 | 584 | **583*** | **18.2!** |
| 20_5_3 | 559 | 557 | 557 | 558 | **554*** | **95!** |
| 20_5_4 | 552 | 549 | 548 | 551 | 548 | **187!** |
| 20_5_5 | 526 | 526 | 526 | 526 | 526 | **58!** |
| 20_5_6 | 513 | 511 | 511 | 511 | 511 | 11 |

(*Continued*)

**Table 5**. (Continued)

| Instances | UB | CSA | SA | HEA | our CP | |
|---|---|---|---|---|---|---|
| | | | | | $C_{max}$ | CPU(s) |
| 20_5_7 | 678 | 674 | 677 | 677 | **672***  | **156!** |
| 20_5_8 | 517 | 516 | 516 | 516 | **513*** | **210!** |
| 20_5_9 | 681 | 679 | 679 | 679 | 679 | 232 |
| 20_5_10 | 525 | 523 | 523 | 525 | **522*** | **114!** |
| 20_10_1 | 797 | 797 | 793 | 796 | **789*** | **670!** |
| 20_10_2 | 844 | 842 | 843 | 844 | **839*** | 465 |
| 20_10_3 | 858 | 858 | 856 | 851 | **846*** | **215!** |
| 20_10_4 | 1015 | 1011 | 1014 | 1015 | **1009*** | **652!** |
| 20_10_5 | 973 | 972 | 973 | 973 | **971*** | 245 |
| 20_10_6 | 796 | 793 | 796 | 794 | **792*** | 639 |
| 20_10_7 | 771 | 771 | 768 | 770 | **766*** | 576 |
| 20_10_8 | 950 | 948 | 947 | 949 | **943*** | 431 |
| 20_10_9 | 953 | 952 | 950 | 948 | **945*** | 116 |
| 20_10_10 | 866 | 862 | 864 | 866 | 862 | 80 |
| 20_15_1 | 1067 | 1066 | 1066 | 1064 | **1062*** | **51!** |
| 20_15_2 | 1333 | 1328 | 1333 | 1327 | **1324*** | 75 |
| 20_15_3 | 1295 | 1295 | 1295 | 1293 | **1292*** | 550 |
| 20_15_4 | 1031 | 1031 | 1031 | 1031 | 1031 | 97 |
| 20_15_5 | 1015 | 1013 | 1013 | 1013 | 1013 | 96 |
| 20_15_6 | 1277 | 1277 | 1277 | 1277 | 1277 | **21.5!** |
| 20_15_7 | 1274 | 1274 | 1274 | 1270 | **1268*** | 625 |
| 20_15_8 | 1261 | 1259 | 1261 | 1261 | **1255*** | 32 |
| 20_15_9 | 1748 | 1748 | 1748 | 1732 | **1721*** | **385!** |
| 20_15_10 | 967 | 967 | 966 | 961 | **959*** | 485 |
| 20_20_1 | 1332 | 1333 | 1333 | 1326 | **1322*** | 735 |
| 20_20_2 | 1325 | 1324 | 1325 | 1325 | **1320*** | 168 |
| 20_20_3 | 1324 | 1320 | 1321 | **1315*** | 1316 | 213 |
| 20_20_4 | 1580 | 1578 | 1578 | 1577 | **1576*** | 207 |
| 20_20_5 | 1320 | 1315 | 1319 | 1317 | **1307*** | 424 |
| 20_20_6 | 1284 | 1284 | 1282 | 1280 | 1280 | **23!** |
| 20_20_7 | 1632 | 1630 | 1632 | 1627 | **1623*** | 425 |
| 20_20_8 | 1847 | 1838 | 1842 | 1832 | **1814*** | **571!** |
| 20_20_9 | 1302 | 1300 | 1302 | 1289 | 1289 | 434 |
| 20_20_10 | 1202 | 1202 | 1204 | 1202 | **1197*** | 154 |

Figs 12 and 13 present the optimal scheduling Gantt charts of the two instances (namely instances 15_15_7 and 20_20_8) with the greatest improvement. The results of these two instances were reduced by 12 and 18 units of time, respectively, from the best-known solution. Moreover, the best feasible solutions for both instances are proved to be optimal for the first time.

## 5.2 Experiment 2: Large scale

To further verify and explore the performance of the proposed IPMMPO-CP on large-scale HFS problems, this subsection conducts comparative experiments on 40 benchmark instances. These 40 large-scale instances can be divided into two categories according to their published sources, among which 10 are from [4], the other 30 are from [9].

Comparative experiments are first conducted on 10 difficult instances published by [4] (see Table 6). Since its release in 2012, the best feasible solutions of these 10 benchmarks have been continuously improved by various well-known heuristics. The representative comparison

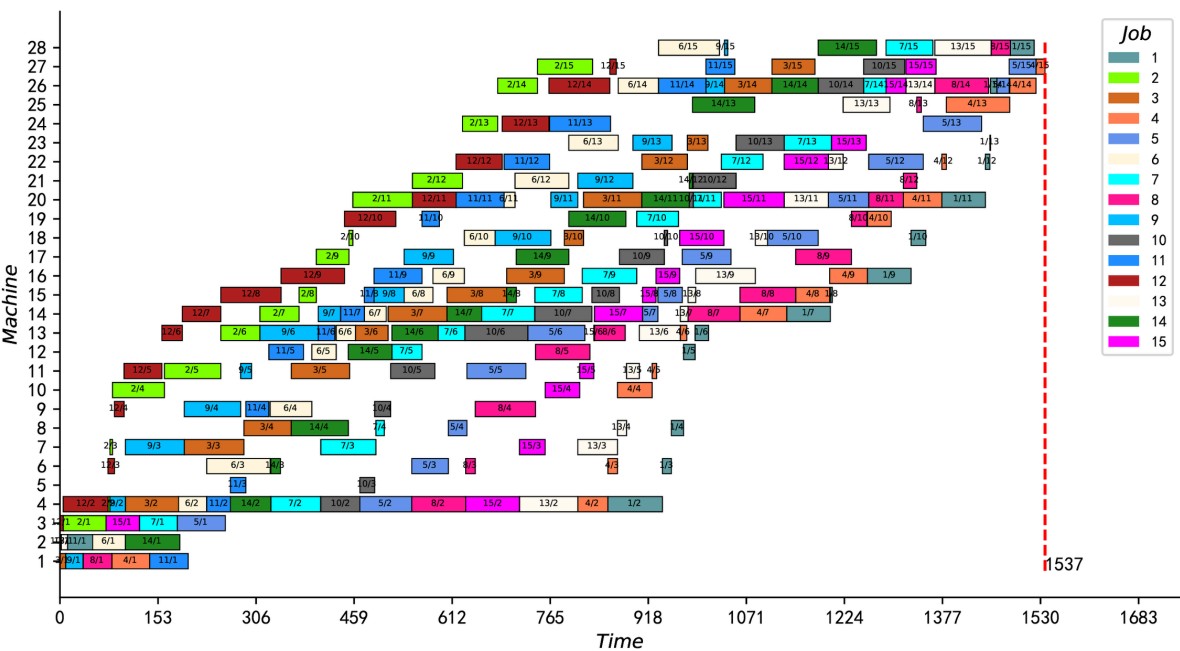

**Fig 12. The optimal scheduling Gantt chart of instance 15_15_7.**

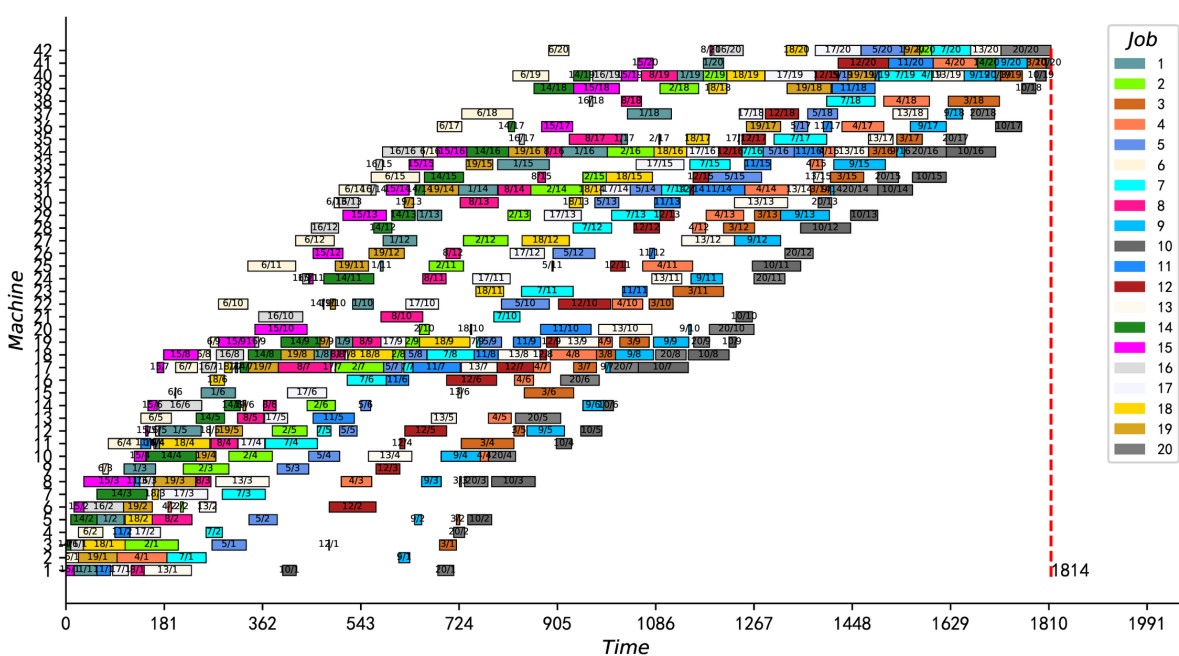

**Fig 13. The optimal scheduling Gantt chart of instance 20_20_8.**

algorithms involved are DABC [21], IDABC [69], and HEA [7]. All the three compared algorithms gave their best solutions on these 10 instances. To ensure the authenticity and accuracy of the data, all comparative data are derived from the literature itself.

**Table 6. Performance comparison of 10 large-scale instances from [4].**

| Instances | DABC | IDABC | HEA | our CP | |
|---|---|---|---|---|---|
| | | | | $C_{max}$ | CPU |
| j30c5e1 | 464 | 463 | 462 | **461*** | 1172 |
| j30c5e2 | 616 | 616 | 616 | 616 | **3.9!** |
| j30c5e3 | 596 | 593 | 593 | **592*** | **879!** |
| j30c5e4 | 566 | 565 | 563 | 563 | 1189 |
| j30c5e5 | 603 | 600 | 600 | 600 | **54.16!** |
| j30c5e6 | 603 | 601 | 599 | 599 | **740!** |
| j30c5e7 | 626 | 626 | 626 | 626 | **280!** |
| j30c5e8 | 674 | 674 | 674 | 674 | 167 |
| j30c5e9 | 643 | 642 | 642 | 642 | **645!** |
| j30c5e10 | 575 | 573 | 571 | **570*** | 1050 |

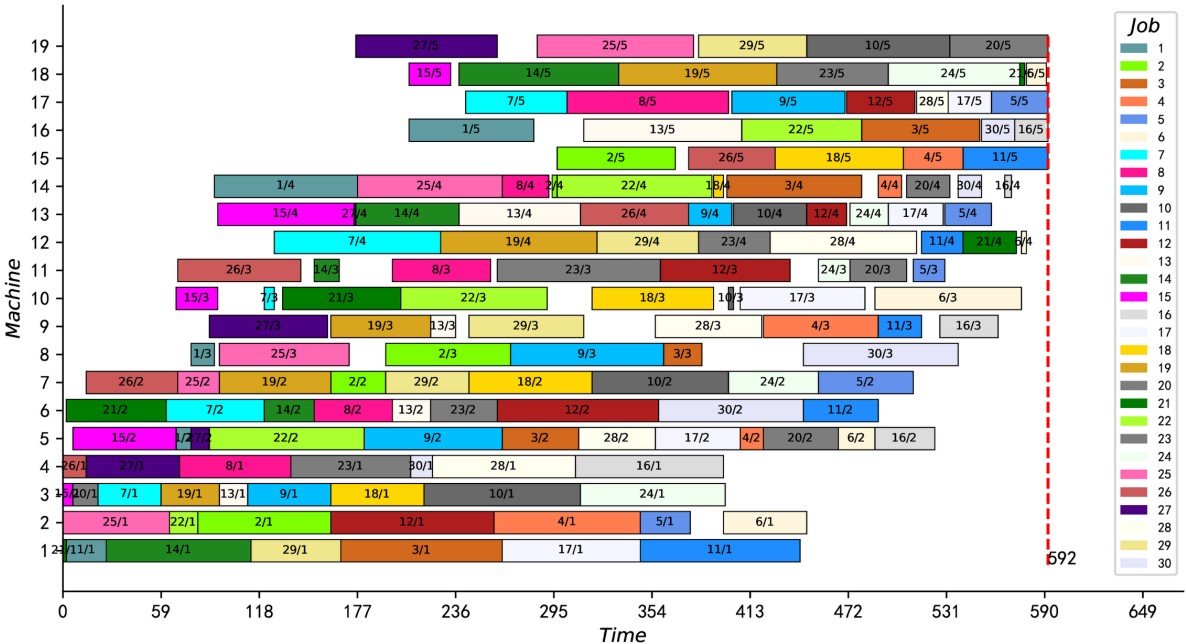

**Fig 14. The optimal scheduling Gantt chart of instance j30c5e3.**

As can be seen from Table 6, the record holder of the best feasible solution so far is obtained by the HEA algorithm. However, IPMMPO-CP not only achieves the existing best-known solutions, but also improves the best feasible solutions for 3 of them. More importantly, we also prove for the first time that 6 best feasible solutions out of 10 instances are optimal. When the search does not stop automatically within the specified time (1200s), it means that it is uncertain whether the best feasible solution found is optimal.

Also, because the proposed IPMMPO-CP uses a deterministic, parallel constraint propagation algorithm that produces the same results every run, the best and average values are always equal. Figs 14 and 15 show the scheduling Gantt charts corresponding to the best feasible solution of instance j30c5e3 and j30c5e10, respectively. Fig 16 is a box plot of the four algorithms on instance j30c5e10. Tables 7 and 8 present the ANOVA results and Tukey HSD tests for several comparative algorithms, respectively. It can be clearly seen that the proposed IPMMPO-CP in this work significantly outperforms the other three comparison algorithms.

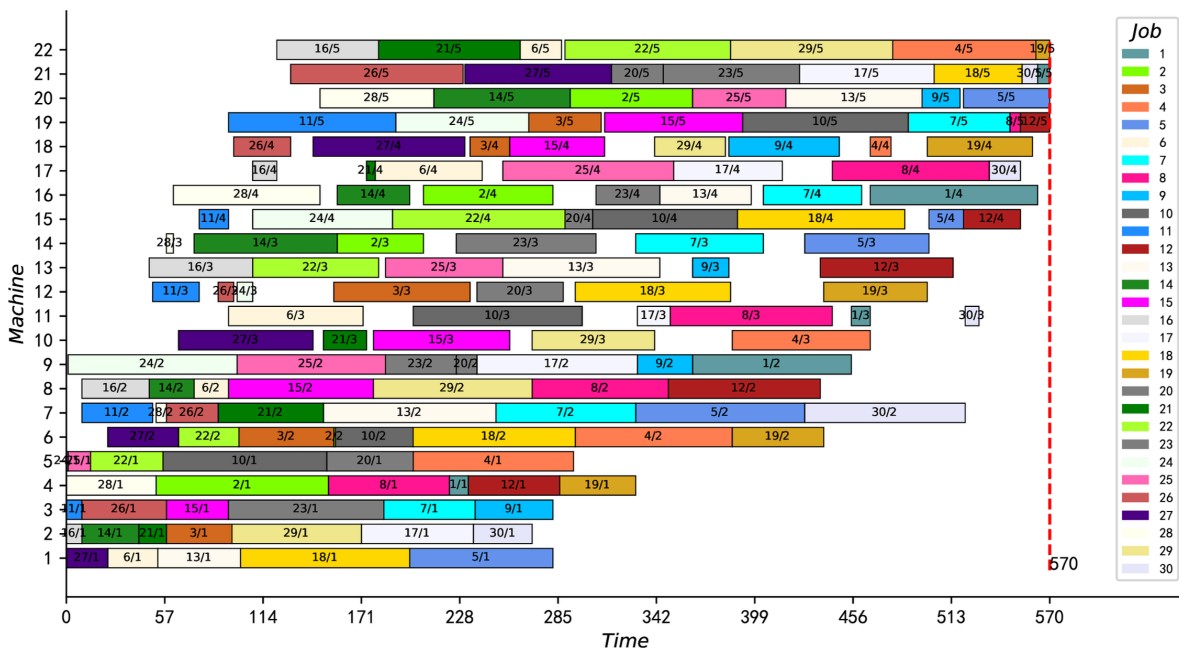

**Fig 15. Scheduling Gantt chart for instance j30c5e10.**

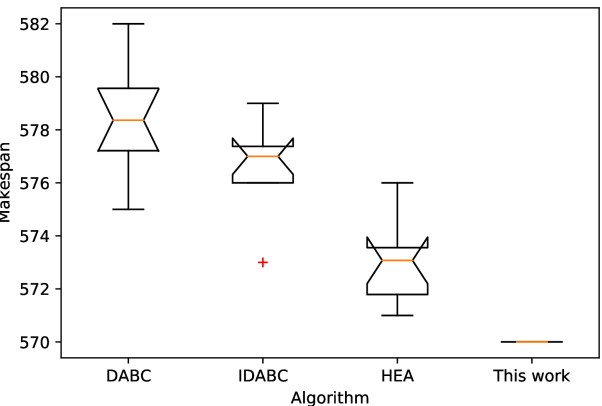

**Fig 16. Box plot comparison on instance j30c5e10.**

**Table 7. ANOVA results from multiple repetitions on large-scale benchmark.**

| Source of Variation | SS | df | MS | F | P-value | F crit |
|---|---|---|---|---|---|---|
| Between Groups | 426.5346 | 3 | 142.1782 | 57.44706 | 8.43E-14 | 2.866266 |
| Within Groups | 89.09794 | 36 | 2.474943 | | | |
| Total | 515.6325 | 39 | | | | |

The following comparative experiments continue on another 30 large-scale instances from [9] (see Table 9). The comparison algorithms involved in Table 9 are improved iterated greedy (IGT) and variable block insertion heuristic (VBIH), both of which were proposed and used

**Table 8. Multiple comparison of means - Tukey HSD.**

| Group1 | Group2 | Meandiff | P-adj | lower | upper | reject |
|--------|--------|----------|-------|-------|-------|--------|
| DABC | HEA | -5.4429 | 0.0000 | -7.3377 | -3.5481 | True |
| DABC | IDABC | -1.7901 | 0.0698 | -3.6849 | 0.1047 | False |
| DABC | our CP | -8.4401 | 0.0000 | -10.3349 | -6.5453 | True |
| HEA | IDABC | 3.6528 | 0.0000 | 1.758 | 5.5476 | True |
| HEA | our CP | -2.9972 | 0.0008 | -4.892 | -1.1024 | True |
| IDABC | our CP | -6.65 | 0.0000 | -8.5448 | -4.7552 | True |

**Table 9. Performance comparison of 30 large-scale instances from [9].**

| Instances | IGT | | VBIH | | $BC_m$ | our CP | |
|-----------|-----|-----|------|-----|--------|--------|-----|
| | Min% [a] | CPU | Min% [a] | CPU | | $C_{max}$ | CPU |
| j40c5e1 | 0.00 | 10.91 | 0.00 | 9.20 | 688 | 688 | **13.2** |
| j40c5e2 | 0.00 | 5.49 | 0.00 | 11.86 | 767 | 767 | **18.6** |
| j40c5e3 | 0.00 | 2.35 | 0.00 | 4.62 | 801 | 801 | **5.4** |
| j40c5e4 | 0.00 | 14.79 | 0.00 | 11.45 | 734 | 734 | **35.7** |
| j40c5e5 | 0.00 | 0.73 | 0.00 | 1.78 | 713 | 713 | **15.3** |
| j40c5e6 | 0.00 | 0.31 | 0.00 | 1.07 | 781 | **780*** | **6.8** |
| j40c5e7 | 0.00 | 3.04 | 0.00 | 3.16 | 735 | 735 | **12.5** |
| j40c5e8 | 0.00 | 14.65 | 0.00 | 13.41 | 802 | 802 | **5.5** |
| j40c5e9 | 0.00 | 1.19 | 0.00 | 3.16 | 840 | 840 | **11.7** |
| j40c5e10 | 0.13 | 18.05 | 0.13 | 16.82 | 774 | 774 | **49.2** |
| j50c5e1 | 0.00 | 13.68 | 0.00 | 24.38 | 970 | 970 | **91.7** |
| j50c5e2 | 0.00 | 7.91 | 0.00 | 11.06 | 994 | 994 | **21.1** |
| j50c5e3 | 0.00 | 21.54 | 0.10 | 14.04 | 1029 | 1029 | 125.0 |
| j50c5e4 | 0.00 | 1.81 | 0.00 | 2.60 | 909 | 909 | **34.3** |
| j50c5e5 | 0.00 | 22.06 | 0.00 | 24.42 | 1026 | 1026 | **113.0** |
| j50c5e6 | 0.00 | 21.75 | 0.00 | 13.85 | 840 | 840 | **86.7** |
| j50c5e7 | 0.00 | 15.84 | 0.00 | 13.93 | 928 | 928 | **14.9** |
| j50c5e8 | 0.00 | 1.93 | 0.00 | 3.17 | 857 | 857 | **10.5** |
| j50c5e9 | 0.15 | 24.37 | 0.15 | 24.65 | 653 | 653 | **76.8** |
| j50c5e10 | 0.00 | 4.83 | 0.00 | 10.71 | 969 | 969 | **8.5** |
| j60c5e1 | 0.00 | 18.44 | 0.00 | 14.83 | 1136 | 1136 | **11.30** |
| j60c5e2 | 0.11 | 28.86 | 0.00 | 23.77 | 920 | 920 | 147.0 |
| j60c5e3 | 0.00 | 1.75 | 0.00 | 3.83 | 1158 | 1158 | **5.3** |
| j60c5e4 | 0.00 | 29.59 | 0.13 | 27.34 | 788 | **787*** | 158.0 |
| j60c5e5 | 0.00 | 10.31 | 0.00 | 20.74 | 1062 | 1062 | **56.6** |
| j60c5e6 | 0.00 | 20.79 | 0.00 | 19.48 | 1048 | **1046*** | **83.2** |
| j60c5e7 | 0.00 | 3.40 | 0.00 | 12.02 | 1225 | 1225 | **3.2** |
| j60c5e8 | 0.00 | 10.69 | 0.00 | 20.76 | 1033 | 1033 | **8.6** |
| j60c5e9 | 0.00 | 2.11 | 0.00 | 13.83 | 1156 | 1156 | **2.6** |
| j60c5e10 | 0.00 | 9.83 | 0.22 | 7.80 | 921 | 921 | **87.3** |
| Avg | 0.01 | 11.43 | 0.02 | 12.79 | | | 44.0 |

[a]Calculation results published in the original literature, which are obtained from Eq (47).

by [9] when publishing these 30 large-scale instances. All comparative data in Table 9 are publicly accessible from the literature [9].The comparative data is presented in the form of relative percentage deviation (RPD), which is calculated as follows.

$$RPD = \frac{BFS - BC_m}{BC_m} \times 100\% \tag{47}$$

where BFS is the best feasible solution found by the algorithms IGT and VBIH, and $BC_m$ is the currently known best solution published in the literature [9]. The time limit of IPMMPO-CP is set to 200s. The "$C_{max}$" column is the best feasible solution found by IPMMPO-CP.

In Table 9, all CPU times are in seconds. The bold data in the last column means that IPMMPO-CP can automatically stop the search, which indicates that the best feasible solution found is also the optimal solution for the corresponding instance. It can be seen that IPMMPO-CP proves for the first time that the current best feasible solution for 27 instances is optimal within the specified time, except for only three instances (i.e., j50c5e3, j60c5e2, and j60c5e4). Although different from the computing environment used for the comparison data, the computation time consumed by IPMMPO-CP is well within acceptable limits, with an average CPU time of 44 seconds.

In addition, we also updated the current best feasible solutions for three instances, see bold data with asterisks in Table 9, and two of them (j40c5e6 and j60c5e6) proved to be optimal. Taking "j40c5e6" as an example, when the CPU time is 1.25s, the feasible solution 781 published in the literature [9] has been searched. Then, the search converges to 780 at 6.8s and stops automatically. Finally, Fig 17 presents the optimal scheduling Gantt chart for "j60c5e6". In conclusion, the experiments in this subsection show that the proposed IPMMPO-CP is efficient and excellent in solving large-scale HFS-IPM problems.

In this section, we mainly compare our approach with nine representative approximation methods. The experimental results demonstrate that IPMMPO-CP outperforms the competing algorithms in both solution time and solution accuracy. This advantage is attributed

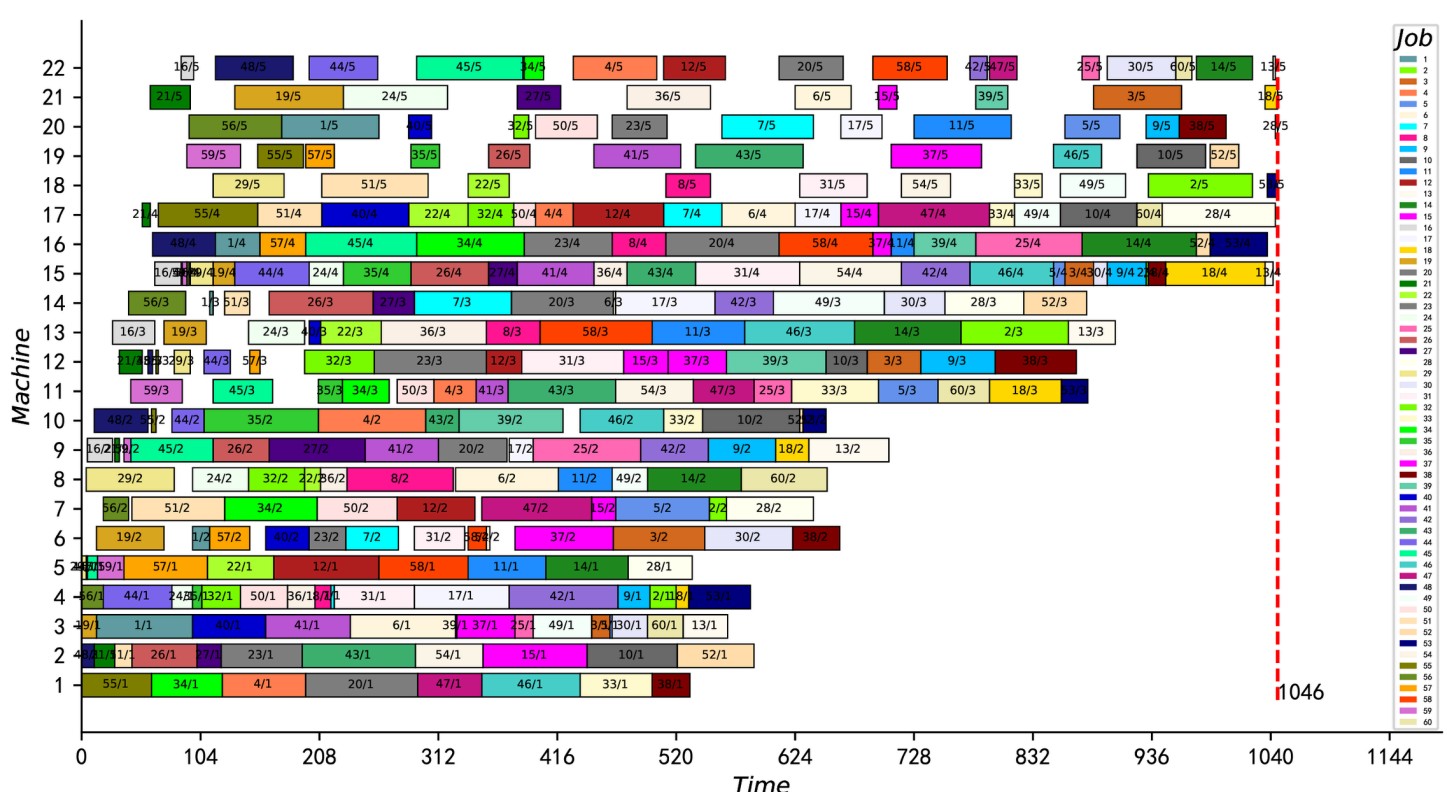

**Fig 17. Gantt chart of an optimal scheduling solution for j60c5e6.**

to the CP model's use of efficient constraint propagation algorithms and domain reduction techniques as its search and inference engine, as well as its support for multi-core and multi-threaded parallel optimization. Not only that, CP methods can even prove that the best feasible solution found is optimal when the search automatically terminates [59,64].

## 6 Application comparison based on HFS-UPM instances and real cases

In real industrial manufacturing, HFS-UPM is closer to the actual production and has more practical significance than HFS-IPM. This section aims to explore the performance of IPMMPO-CP in solving HFS-UPM type problems. We compare IPMMPO-CP with two existing CP models on several instances and real cases. All instances and cases involved in this section belong to the HFS-UPM type. The time limit for the IPMMPO-CP model is set to 600s to ensure consistency with the compared CP models.

First, the proposed IPMMPO-CP is compared with CP1 [71] in the no-wait HFS scenario, involving 4 instances from reference [74]. To ensure the validity of the comparison, we fine-tune a constraint of IPMMPO-CP as follows. The constraint Eq (44) in the Sect 4.2 is replaced by the following Eq (48) to make it applicable for the no-wait HFS scenario. The no-wait constraint can be satisfied by replacing the keyword "endBeforeStart" with "endAtStart", that is, each job must be processed continuously in all stages without interruption. Table 10 shows the comparative analysis with model CP1 on 4 instances in the no-wait HFS scenario.

$$endAtStart\ (PRO[o_1], PRO[o_2]),$$
$$\forall i \in N, o_1, o_2 \in P_{OId,i,h} : o_1.i = o_2.i\ \&\ 1 + o_1.h = o_2.h \tag{48}$$

The bold data in the "Best" column of Table 10 represents the optimal solution found by CP1 and IPMMPO-CP in the corresponding "CPU" solution time. The data from Table 10 indicates that both compared CP models can find the optimal solution to the problem within the specified time. However, it is evident that CP1 requires higher time costs.

Second, the proposed IPMMPO-CP is compared with CP2 [72] on 2 instances and 2 real cases. The comparison here is divided into two categories: the standard HFS scenario and the no-wait HFS scenario. The instances and real cases involved in this comparison are all from the literature [10]. Table 11 shows the comparative analysis with model CP2 in standard HFS scenario. As both models involve the same number of variables, Table 11 shows a comparison in terms of the constraint count "#Cstr", the best solution "Best", and the solving time "CPU". RealCase1~2 are two real cases from a water meter manufacturer. The last RealCase2 is a large-scale case involving 50 jobs, 22 machines, and 6 stages. The following conclusions can be drawn from the comparative data in Table 11.

**Table 10. Comparison with CP1 [71] in the no-wait HFS scenario.**

| Instance | CP1 | | IPMMPO-CP | |
|---|---|---|---|---|
| | Best | CPU | Best | CPU |
| 1 | **21** | 6.2 | **21** | 0.8 |
| 2 | **23** | 0.9 | **23** | 0.25 |
| 3 | **297** | 600 | **297** | 51.83 |
| 4 | **42** | 75.4 | **42** | 53.96 |
| Average | | 170.62 | | 26.71 |

Table 11. Comparison with CP2 [72] in the standard HFS scenario.

| Instance | CP2 | | | IPMMPO-CP | | |
|---|---|---|---|---|---|---|
| | Best | #Cstr | CPU[a] | Best | #Cstr | CPU[a] |
| BenCase1 | **23** | 175 | 0.39 | **23** | 76 | 0.25 |
| BenCase2 | **297** | 212 | 0.23 | **297** | 102 | 0.18 |
| RealCase1 | **479** | 212 | 16.93 | **479** | 102 | 3.5 |
| RealCase2 | 1901 | 1660 | 489.26 | 1887 | 582 | 384.75 |
| Average | | | 126.70 | | | 97.17 |

[a]The first time the CP solver finds the optimal/best solution.

- Both CP models achieved the optimal solution in the first three cases. The IPMMPO-CP model updated the current best feasible solution for RealCase2. In all cases, the computational time used by IPMMPO-CP is less than that of CP2.
- From the perspective of model constraint count "#Cstr", it is evident that IPMMPO-CP is more concise than CP2. This is also a crucial factor contributing to the lower computational time cost required by IPMMPO-CP.

Table 12 shows the comparative analysis with model CP2 on the same instances in the no-wait HFS scenario. Compared with standard HFS, the number of variables and constraints in the no-wait HFS scenario have not changed. IPMMPO-CP obtains the optimal solutions for the first three cases, and the time consumption is less than CP2. In addition, IPMMPO-CP also updates the current best feasible solution of RealCase2 in the no-wait scenario. The comparison results indicate that the proposed IPMMPO-CP model has fewer constraints and requires shorter computational time. Fig 19 and Fig 18 give the scheduling Gantt charts corresponding to the best feasible solutions of RealCase2 in the standard and no-wait HFS scenario, respectively.

In this section, we mainly compare our approach with two recent and representative CP models, namely CP1 and CP2. The CP1 model lacks cumulative constraints, which may result in a less compact search space and consequently higher computational time. Although the CP2 model employs cumulative constraints, they are fundamentally different from those used in our study. As shown in Tables 11 and 12, the number of constraints in the CP2 model (see the "#Cstr" indicator) is 2–3 times greater than that of IPMMPO-CP, which significantly increases the model's complexity. In summary, in terms of solution time, solution accuracy, and model compactness, IPMMPO-CP outperforms both CP1 and CP2.

Table 12. Comparison with CP2 [72] in the no-wait HFS scenario.

| Instance | CP2 | | | IPMMPO-CP | | |
|---|---|---|---|---|---|---|
| | Best | #Cstr | CPU[a] | Best | #Cstr | CPU[a] |
| BenCase1 | **23** | 175 | 1.49 | **23** | 76 | 0.3 |
| BenCase2 | **297** | 212 | 29.08 | **297** | 102 | 2.65 |
| RealCase1 | **485** | 212 | 7.65 | **485** | 102 | 1.72 |
| RealCase2 | 1966 | 1660 | 542.63 | 1955 | 582 | 487.36 |
| Average | | | 145.21 | | | 123.01 |

[a]The first time the CP solver finds the optimal/best solution.

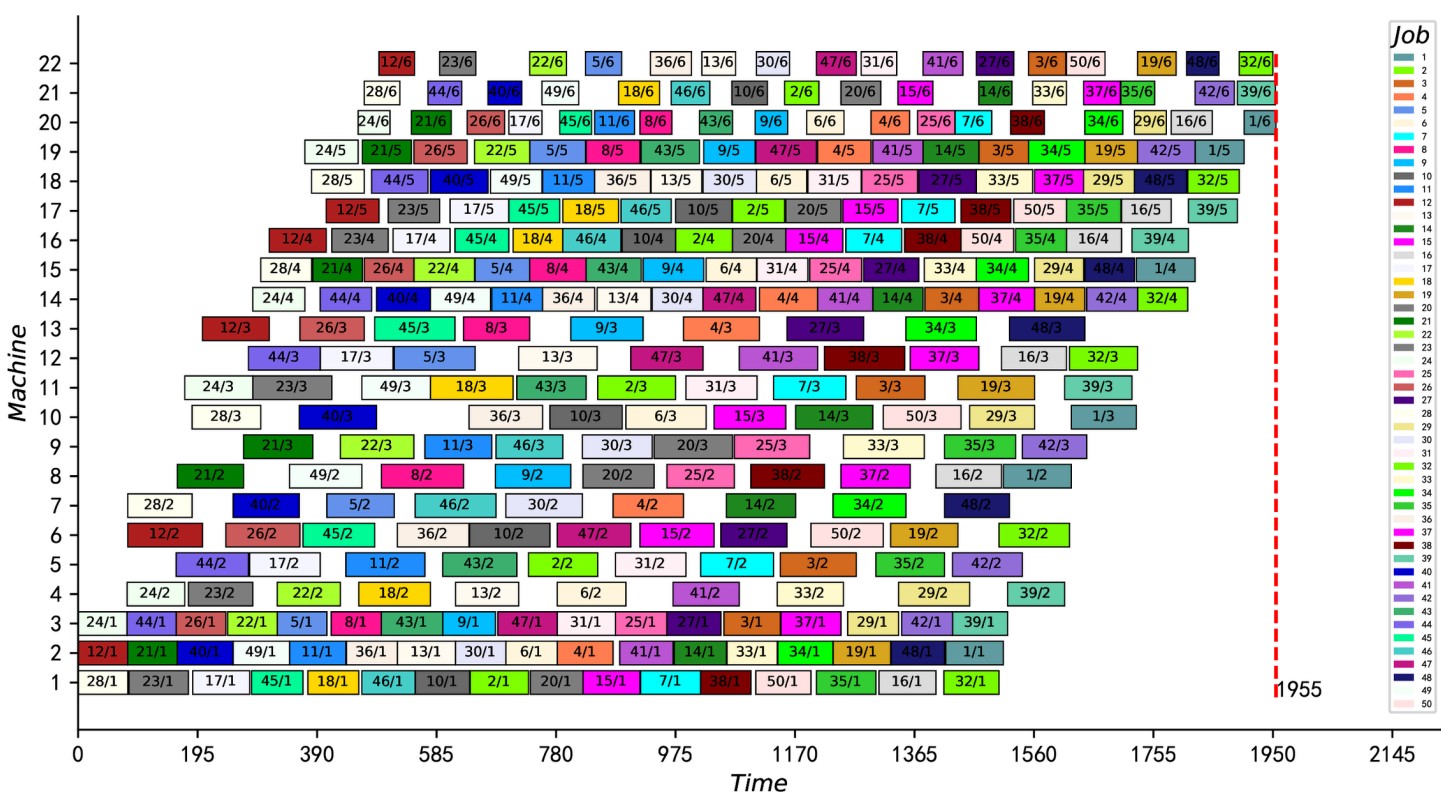

**Fig 18. Gantt chart corresponding to the best feasible solution of RealCase2 in the no-wait HFS scenario.**

## 7 Conclusions

This work explores the application research of the probe machine in the field of scheduling for the first time. First, considering the problem characteristics of HFS and the computational advantages of PM, an improved PM model allowing multi-level probe operations is proposed. This model can universally address both HFS-IPM and HFS-UPM problems. Second, inspired by the data libraries of IPMMPO, two tuple sets were further constructed, laying the foundation for the subsequent construction of the IPMMPO-CP model. In order to verify and explore the performance of the IPMMPO-CP model, we first conducted a large number of experimental comparisons with 9 representative algorithms on HFS-IPM problems. The results indicate that the IPMMPO-CP method outperforms the comparison algorithms on instances of various sizes. Subsequently, we compared it with the two latest CP models on HFS-UPM problems. These comparisons include benchmark instances, real cases, the standard HFS scenario, and the no-wait HFS scenario. The results show that the proposed IPMMPO-CP outperforms the two compared CP models.

**Practical deployment considerations:** While IPMMPO-CP demonstrates computational advantages, its industrial implementation requires: (a) Reliable sensor networks for real-time job tracking (e.g., RFID/IIoT devices); (b) Edge computing infrastructure to handle data collection delays; (c) Middleware integration with existing MES/ERP systems via OPC-UA or API gateways.

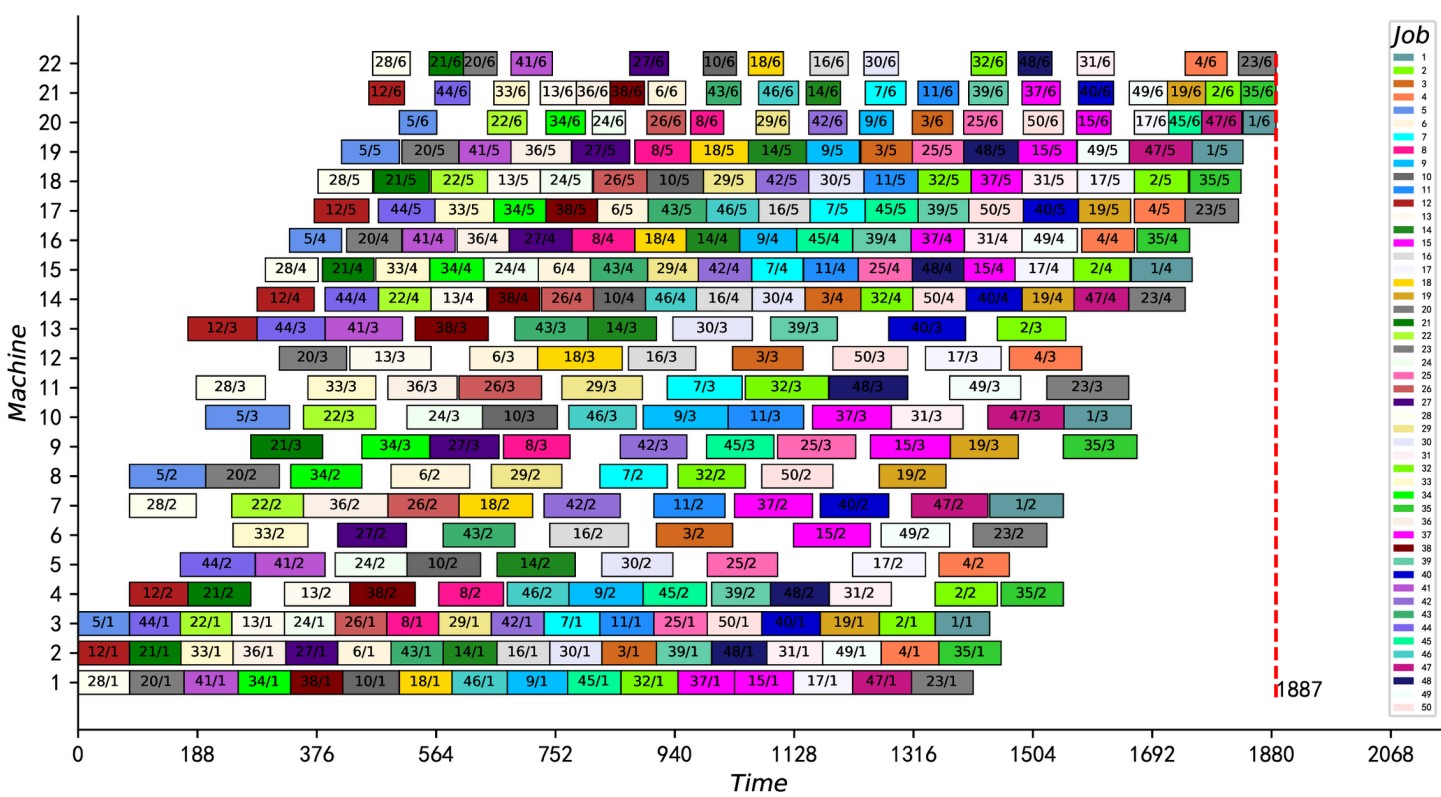

**Fig 19. Gantt chart corresponding to the best feasible solution of RealCase2 in the standard HFS scenario.**

The existing shortcomings of this study, which can also be used as directions for continuous improvement in follow-up work, are summarized as follows. (1) In CP modeling, a question often troubles us, that is, is there a measurable trade-off between the number of constraints in the model and its solving efficiency? How to evaluate and balance this contradiction will become an important guide for studying the compactness and effectiveness of CP models. (2) In addition to the OPL keywords involved in this study, do other keywords help improve the compactness and effectiveness of the CP model? (3) The proposed IPMMPO-CP only involves single-objective optimization. However, under the needs of the green and low-carbon era, future research can consider multi-objective CP modeling of green scheduling problems with energy-saving and emission reduction. (4) We fully acknowledge the importance of computational efficiency in solving large-scale, multi-scenario problems. In future work, we plan to integrate the Rolling Horizon Decomposition (RHD) approach to further enhance the timeliness of the proposed method.

## Supporting information

**S1 File.**
(RAR)

## Acknowledgments

We would like to thank Professor Jin Xu from Peking University for pioneering research in the field of DNA computing.

## Author contributions

**Data curation:** Xiang Tian.

**Funding acquisition:** Yang Kong, Xiyu Liu.

**Methodology:** Xiang Tian.

**Project administration:** Xiyu Liu.

**Resources:** Xiang Tian.

**Supervision:** Yang Kong, Xiyu Liu.

**Validation:** Yang Kong, Xiyu Liu.

**Visualization:** Xiang Tian.

**Writing – original draft:** Xiang Tian.

**Writing – review & editing:** Xiang Tian.

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
