## [Decision Letter · Decision Letter 0]

31 May 2025

PONE-D-25-15681Solving Multi-Scenario Hybrid Flow Shop Scheduling Problem Based on an Improved Probe Machine ModelPLOS ONE

Dear Dr. Tian,

Thank you for submitting your manuscript to PLOS ONE. After careful consideration, we feel that it has merit but does not fully meet PLOS ONE’s publication criteria as it currently stands. Therefore, we invite you to submit a revised version of the manuscript that addresses the points raised during the review process.

We look forward to receiving your revised manuscript.

Kind regards,

Shih-Wei Lin

Academic Editor

PLOS ONE

Journal Requirements:

Journal Requirements:

We would like to thank Professor Jin Xu from Peking University for pioneering research

in the field of DNA computing. This work was supported in part by the Social Science

Plannig Project of Shandong under Grant 22CGLJ01, in part by the National Natural

Science Foundation of China under Grant 72274023, in part by the Natural Science

Foundation of Shandong Province under Grant ZR202112020423.

1. The Social Science Plannig Project of Shandong under Grant 22CGLJ01;

2. The National Natural Science Foundation of China under Grant 72274023;

3. The Natural Science Foundation of Shandong Province under Grant ZR202112020423.

The recipient of all three funding grants mentioned above is the author, Yang Kong.

Professor Kong is in charge of fund acquisition, project management and supervising the rationality of this work.

Additional Editor Comments:

The reviewers have commented on your above paper. They indicated that it is not acceptable for publication in its present form.

However, if you feel that you can suitably address the reviewers' comments , I invite you to revise and resubmit your manuscript.

Please carefully address the issues raised in the comments.

Reviewers' comments:

Reviewer's Responses to Questions

**Comments to the Author**

1. Is the manuscript technically sound, and do the data support the conclusions?

Reviewer #1: Yes

Reviewer #2: No

2. Has the statistical analysis been performed appropriately and rigorously? 

Reviewer #1: Yes

Reviewer #2: Yes

3. Have the authors made all data underlying the findings in their manuscript fully available?

Reviewer #1: Yes

Reviewer #2: No

4. Is the manuscript presented in an intelligible fashion and written in standard English?

Reviewer #1: Yes

Reviewer #2: Yes

5. Review Comments to the Author

Reviewer #1: Authors used an improved probe machine model for solving a multi-scenario hybrid flow shop scheduling problem. It is very interesting and well written. However, some questions need to be solved as follwos:

1. How to reflect the multi-scenario hyrbid flow shop problem?

2. Please provide mathematical model of the multi-scenario HFSP. Does multi-scenario HFSP have the same mathematical model or different mathematical models?

3. In introduction, there are various studies which have a close relation with the topic considered in this paper, such as An automatic multi-objective evolutionary algorithm for the hybrid flowshop scheduling problem with consistent sublots, and Mathematical model and knowledge-based iterated greedy algorithm for distributed assembly hybrid flow shop scheduling problem with dual-resource constraints. The review of this paper will be further enriched if they are included.

4. In the experiment, statistic test like t-test should be conducted to validate the significance difference between difference metaheuristics for large-scale problems due to the statistical probability characteristics of these metaheuristics. Moreover, the deep reason why the proposal is better than its counterparts should be stated more clearly in the result analysis.

Reviewer #2: This work tackles the hybrid flow shop scheduling challenge using an improved probe machine model. It introduces a novel approach called IPMMPO, which handles multiple scenarios like identical or unrelated parallel machines, no-wait, and standard cases. They developed new data libraries and formulated a constraint programming model that was tested extensively, outperforming several representative algorithms. Certainly. Based on my analysis of the paper, here are some potential issues or areas for improvement:

1. Although the authors propose the IPMMPO-CP model and demonstrate its effectiveness on several instances, the experimental validation mainly focuses on specific case studies. There may be concerns about the model's generalizability and robustness across various real-world manufacturing scenarios.

2. While the model performs well on medium-sized problems, solving larger or more complex instances—especially in multi-scenario, multi-machine environments—could lead to significant computational time, possibly limiting practical application in large-scale industrial settings.

3. The paper references the design of data libraries and probe libraries essential for the model, but it does not provide detailed insights into their design principles, data completeness, or scalability. Poorly designed libraries could affect model performance and adaptability.

4. Although the model shows promising results in simulations and benchmark instances, integrating such a system into actual manufacturing environments might face challenges such as sensor reliability, real-time data collection, and system integration, which are not discussed in the paper.

5. The impact of data uncertainties or parameter variations on the model’s solution quality and stability is not explored.

6. In Table4, it should add "our CP" above the last two columns, such as like Table3. In addition, it's more clear to add "*" for instance (20_20_3) that found by HEA.

6. PLOS authors have the option to publish the peer review history of their article (what does this mean?). If published, this will include your full peer review and any attached files.

Reviewer #1: No

Reviewer #2: No

---

## [Author Response · Author response to Decision Letter 1]

25 Jun 2025

As figures and tables cannot be submitted and displayed here, please refer to the “Response to Reviewers” document we uploaded to the submission system. In that document, we have provided detailed and comprehensive responses.

---

## [Decision Letter · Decision Letter 1]

25 Jul 2025

Solving multi-scenario hybrid flow shop scheduling problem based on an improved probe machine model

PONE-D-25-15681R1

Dear Dr. Tian,

We’re pleased to inform you that your manuscript has been judged scientifically suitable for publication and will be formally accepted for publication once it meets all outstanding technical requirements.

Kind regards,

Shih-Wei Lin

Academic Editor

PLOS ONE

Additional Editor Comments (optional):

The authors have invested a substantial effort to address all issues indicated in the previous review round for two reviewers, thus significantly improving the quality of their paper. Therefore, I suggest an acceptance.

Reviewers' comments:

Reviewer's Responses to Questions

**Comments to the Author**

1. If the authors have adequately addressed your comments raised in a previous round of review and you feel that this manuscript is now acceptable for publication, you may indicate that here to bypass the “Comments to the Author” section, enter your conflict of interest statement in the “Confidential to Editor” section, and submit your "Accept" recommendation.

Reviewer #1: All comments have been addressed

Reviewer #2: All comments have been addressed

2. Is the manuscript technically sound, and do the data support the conclusions?

Reviewer #1: Yes

Reviewer #2: Yes

3. Has the statistical analysis been performed appropriately and rigorously? 

Reviewer #1: Yes

Reviewer #2: Yes

4. Have the authors made all data underlying the findings in their manuscript fully available?

Reviewer #1: Yes

Reviewer #2: Yes

5. Is the manuscript presented in an intelligible fashion and written in standard English?

Reviewer #1: Yes

Reviewer #2: Yes

6. Review Comments to the Author

Reviewer #1: All my concerned questions have been done. I'm very satisfied with the author's reply. It can be accepted in the current verision.

Reviewer #2: (No Response)

7. PLOS authors have the option to publish the peer review history of their article (what does this mean?). If published, this will include your full peer review and any attached files.

Reviewer #1: No

Reviewer #2: No

---

## [Editor Report · Acceptance letter]

PONE-D-25-15681R1

PLOS ONE

Dear Dr. Tian,

I'm pleased to inform you that your manuscript has been deemed suitable for publication in PLOS ONE. Congratulations! Your manuscript is now being handed over to our production team.

Kind regards,

on behalf of

Professor Shih-Wei Lin

Academic Editor

PLOS ONE